# Altered expression of a quality control protease in *E. coli* reshapes the in vivo mutational landscape of a model enzyme

Samuel Thompson[1]*, Yang Zhang[2], Christine Ingle[3], Kimberly A Reynolds[3,4], Tanja Kortemme[1,2,5]*

[1]Graduate Group in Biophysics, University of California San Francisco, San Francisco, United States; [2]Department of Bioengineering and Therapeutic Sciences, University of California San Francisco, San Francisco, United States; [3]The Green Center for Systems Biology, University of Texas Southwestern Medical Center, Dallas, United States; [4]Department of Biophysics, University of Texas Southwestern Medical Center, Dallas, United States; [5]Chan Zuckerberg Biohub, San Francisco, United States

**Abstract** Protein mutational landscapes are shaped by the cellular environment, but key factors and their quantitative effects are often unknown. Here we show that Lon, a quality control protease naturally absent in common *E. coli* expression strains, drastically reshapes the mutational landscape of the metabolic enzyme dihydrofolate reductase (DHFR). Selection under conditions that resolve highly active mutants reveals that 23.3% of all single point mutations in DHFR are advantageous in the absence of Lon, but advantageous mutations are largely suppressed when Lon is reintroduced. Protein stability measurements demonstrate extensive activity-stability tradeoffs for the advantageous mutants and provide a mechanistic explanation for Lon's widespread impact. Our findings suggest possibilities for tuning mutational landscapes by modulating the cellular environment, with implications for protein design and combatting antibiotic resistance.

**\*For correspondence:**
Thompson.SamuelM@gmail.com (ST);
kortemme@cgl.ucsf.edu (TK)

**Competing interests:** The authors declare that no competing interests exist.

## Introduction

Natural protein sequences are constrained by pressures to maintain required structures and functions within a complex cellular environment. However, key cellular factors shaping protein sequences (such as interactions with cellular binding partners or with the proteostasis machinery) are often unknown. To characterize functional constraints, it has been useful to determine mutational landscapes of proteins, which we define here as the effects on growth of every possible single amino acid mutation in the protein, via deep mutational scanning (*Boucher et al., 2016*; *Fowler and Fields, 2014*). Deep mutational scanning studies have provided insights into evolution of new protein functions (*McLaughlin et al., 2012*; *Stiffler et al., 2015*; *Wrenbeck et al., 2017*), protein design (*Tinberg et al., 2013*; *Whitehead et al., 2012*), functional trade-offs (*Klesmith et al., 2017*; *Steinberg and Ostermeier, 2016*), and adaptation to altered environments (*Hietpas et al., 2013*). With a few exceptions (*Bandaru et al., 2017*; *Hietpas et al., 2013*; *Jiang et al., 2013*; *Stiffler et al., 2015*), however, these studies find a general tolerance to mutation for residues outside of active sites and binding interfaces (*Araya et al., 2012*; *Boucher et al., 2016*; *Klesmith et al., 2017*; *Roscoe et al., 2013*; *Wrenbeck et al., 2017*) that is often explained by the absence of key environmental constraints under the selection conditions (*Bandaru et al., 2017*; *Jiang et al., 2013*; *Stiffler et al., 2015*).

To study the impact of multiple constraints on mutational tolerance during selection, we chose *E. coli* dihydrofolate reductase (DHFR) as a model system. DHFR is an essential enzyme within folate

metabolism that reduces dihydrofolate to tetrahydrofolate and is necessary for thymidine production. Using this activity as the basis for an in vivo selection assay (*Reynolds et al., 2011*), we aimed first to measure a mutational landscape for DHFR and then to determine how a change to the cellular environment might affect the landscape. Because DHFR is known to progress through multiple conformational states during catalysis (*Boehr et al., 2006*; *Sawaya and Kraut, 1997*; *Figure 1—figure supplement 1*), we expected the mutational landscape of DHFR to be constrained by the requirement to adopt these different conformations. Moreover, prior work had suggested DHFR is impacted by cellular constraints such as protein quality control (*Bershtein et al., 2013*) and the build-up of a toxic metabolic intermediate (*Schober et al., 2019*). We hence expected deep mutational scanning to reveal a highly constrained mutational landscape for DHFR that would contrast with the mutational tolerance observed in other systems.

## Results

As the basis for our studies, we first sought to establish highly sensitive selection conditions for DHFR function that would be calibrated to DHFR enzymatic velocity (rate of DHF conversion per molecule of DHFR) and capable of resolving mutants with velocities near-to or faster-than wild-type. We anticipated that we would need to control DHFR protein expression (intracellular abundance) levels because two prior studies that modified the chromosomal DHFR gene had reported an overall high mutational tolerance under permissive selection conditions (*Garst et al., 2017*) and that DHFR abundance can be reduced to ~30% without a growth impact (*Bershtein et al., 2013*). We used an *E. coli* strain derived from ER2566 with the genes for DHFR and a downstream enzyme, thymidylate synthase, deleted in the genome and complemented on a pACYC-DUET plasmid with a weak ribosome binding site (see Materials and methods) that results in DHFR abundance at approximately 10% of the endogenous protein level (*Figure 1—figure supplement 2*, *Figure 1—source data 1*). To tightly control growth conditions, we performed selections in a turbidostat to maintain the culture in early Log phase growth (*Figure 1A*, *Figure 1—figure supplement 3A*). To quantify the effects of DHFR mutations on growth, we calculated selection coefficients (*Rubin et al., 2017*) from the change in allele frequency over time by deep sequencing of timepoint samples determined in biological triplicate (*Figure 1B*). For a panel of 14 DHFR mutants, we confirmed that the selection coefficients obtained from deep mutational scanning correlated linearly with growth rates measured separately for the individual variants in a plate reader (*Figure 1—figure supplement 3B*, *Figure 1—source data 2*), as expected. Furthermore, under our controlled selection conditions, we observed a linear relationship between selection coefficient and in vitro velocity (*Figure 1C*) at cytosolic substrate concentrations (*Bennett et al., 2009*; *Kwon et al., 2008*) for these DHFR mutants (*Figure 1—source data 3*). These results confirm that selection coefficients between −1.5 and 1.0 in our experiment are correlated with DHFR enzymatic velocity over approximately 3 orders of magnitude, and that selection can resolve mutants with higher velocities than wild-type level velocity.

We next analyzed the deep mutational scanning data for all possible DHFR single point mutants under the calibrated selection conditions (*Figure 1D*, *Supplementary file 1*). All pairwise replicates were related with a Pearson correlation $R^2$ value of 0.70 and the median standard deviation between replicates for all selection coefficients was 0.2 (*Figure 1—figure supplement 3C–E*). Using this value, we defined the selection coefficient interval of 0 ±0.2 as WT-like behavior. Within this interval, the standard deviation of the selection coefficients between replicates was not correlated with changes in selection coefficient (*Figure 1—figure supplement 4A*). Moreover, our WT-like threshold of 0.2 was greater than the value of 0.12 for the standard deviation for wild-type synonymous codons (*Figure 1—figure supplement 4B*). Based on these considerations, we defined DHFR mutations with selection coefficients of <−0.2 and >0.2 as disadvantageous and advantageous, respectively. Mutations that were depleted during overnight growth (under less stringent conditions using a supplemented growth medium, see Materials and methods) were assigned a null phenotype. As expected, mutations at DHFR positions that are known to be functionally important (M20, W22, D27, L28, F31, T35, M42, L54, R57, T113, G121, D122, and S148) were generally disadvantageous or null mutations (*Figure 1—figure supplement 5*). These results indicate that our selection assay is a sensitive reporter of functionally important residues and that our results are consistent with previous biochemical characterization of DHFR.

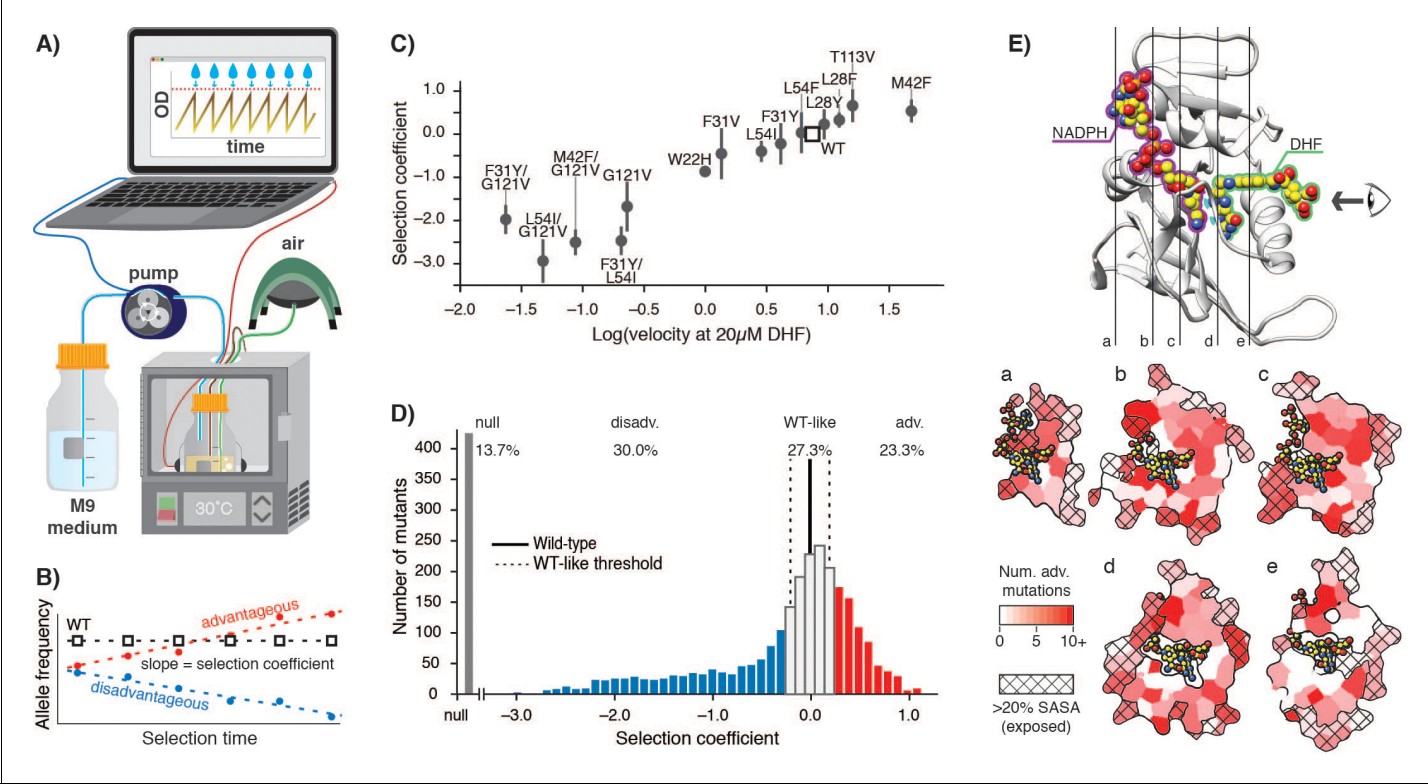

**Figure 1.** *E. coli* DHFR deep mutational scanning uncovers many advantageous mutations. (**A**) Turbidostat schematic. Reoccurring dilutions with fresh medium keep the culture optical density (OD600) below 0.075. (**B**) The selection coefficient for each mutant is the slope of the linear regression of allele frequency over time. The wild-type (squares) value is normalized to zero. Advantageous (red) mutations increase and disadvantageous (blue) mutations decrease in frequency. (**C**) Selection coefficients from deep mutational scanning as a function of enzymatic velocity for purified DHFR point mutants measured in vitro. Velocities at 20 μM DHF were calculated from Michaelis-Menten parameters. Error bars reflect the standard deviation from three biological replicates. (**D**) Histogram of selection coefficients. The wild-type value is indicated with a vertical black line. The median standard deviation over all mutations is the cut-off for WT-like behavior (Materials and methods, *Figure 1—figure supplement 3*, *Figure 1—figure supplement 4*) and is indicated with dashed lines. Mutation are colored as advantageous (red), disadvantageous (blue), WT-like (white), or null (grey). (**E**) Structural model of DHFR (PDB ID: 3QL3) with cross-section slices (a–e) indicated. The DHF substrate (green) and the NADPH cofactor (purple) are represented by spheres (yellow carbons and heteroatom coloring). An arrow indicates the perspective for each slice. (a–e) five cross-section slices. Color scale indicates numbers of advantageous mutations at each position. Crosshatching indicates residues with >20% solvent accessible surface area.

The online version of this article includes the following source data and figure supplement(s) for figure 1:

**Source data 1.** Soluble DHFR expression levels in molecules per cell measured from lysate activity assays as described in Materials and methods.

**Source data 2.** Selection coefficients for –Lon selection (*Figure 1—source data 1*) compared to monoculture growth rates measured in a plate reader in *ER2566 ΔfolA/ΔthyA (–Lon)* as described in Materials and methods.

**Source data 3.** Michaelis-Menten kinetics for the set of DHFR mutants (*Fierke and Benkovic, 1989*; *Huang et al., 1994*; *Reynolds et al., 2011*) used to calibrate the selection are reported together with the reference from which the values were taken.

**Figure supplement 1.** Conformations adopted during the DHFR catalytic cycle: 1RX1, 3QL3, 1RX4, and 1RX5 and a QMMM model of the hydride transfer step (*Liu et al., 2013*) represent the conformational states adopted by DHFR over the catalytic cycle.

**Figure supplement 2.** Soluble WT DHFR cellular abundance for endogenous (chromosomal) DHFR in the parental strain and DHFR expressed from plasmids in the selection system.

**Figure supplement 3.** Determination of selection coefficients for DHFR.

**Figure supplement 4.** Variation in selection coefficients for –Lon selection.

**Figure supplement 5.** Residues previously known to have a functional role shown on the DHFR structure.

**Figure supplement 6.** Growth curves for top advantageous mutations.

**Figure supplement 7.** Example positions with multiple advantageous mutations hypothesized to be destabilizing, shown on the DHFR structure.

In previous deep mutational scanning experiments, stringent selection typically revealed many disadvantageous mutations (*Garst et al., 2017*; *Jiang et al., 2013*; *Mavor et al., 2016*; *Mavor et al., 2018*; *Stiffler et al., 2015*). In contrast, the most striking observation under our conditions is the large fraction of advantageous mutations (red, *Figure 1D*): 736 of 3161 possible variants

were advantageous (23.3%), and wild-type DHFR only ranked 1203[rd] (although 467 of the 1202 higher-ranking variants fall into the WT-like interval). In direct measurements of individual growth rates under our selection conditions, the top two DHFR variants (W47L and L24V) led to increases in growth rate of 40% and 76%, respectively, when compared to wild-type DHFR (*Figure 1—figure supplement 6*). Advantageous mutations were widely distributed over 127 of the 159 positions of DHFR (*Figure 1E*). Furthermore, when we examined the DHFR structure, many of the advantageous mutations appeared to disrupt key side-chain interactions, for example by disrupting atomic packing interactions or surface salt-bridges (*Figure 1—figure supplement 7*).

To understand the origins of this counter-intuitive prevalence of advantageous mutations, we looked for cellular factors potentially affecting our mutational landscape. Our selection strain (*Anton et al., 2016*), like most standard expression strains of *E. coli*, is naturally deficient in Lon protease (*Gur and Sauer, 2008*) due to an insertion of IS186 in the *lon* promoter region (*saiSree et al., 2001*). Lon is a major component of protein quality control in *E. coli* (*Powers et al., 2012*; *Sauer and Baker, 2011*) responsible for degrading poorly folded proteins. Moreover, Lon had previously been implicated in degrading DHFR unstable variants in *E. coli* (*Bershtein et al., 2013*; *Cho et al., 2015*), and deleting Lon in an MG1655 strain of *E. coli* masked the deleterious impact of 2 destabilizing mutations out of a panel of 21 mutants tested in growth experiments at 30 ˚C (*Bershtein et al., 2013*). Although these 21 mutants were selected for minimal impacts on Michaelis-Menten kinetic parameters, we reasoned that the absence of Lon could be responsible for the large fraction of advantageous but potentially destabilizing mutations observed in our selection.

To test this prediction, we reintroduced chromosomal Lon expression under the control of a constitutive promoter in our selection strain, and repeated deep mutational scanning in biological triplicate (*Supplementary file 2*). We refer to the two regimes as +Lon and –Lon selection. The quality of +Lon selection was comparable to that of –Lon selection (*Figure 2—figure supplement 1*, *Figure 2—figure supplement 2*). Consistent with our hypothesis, the distribution of selection coefficients shifted towards more negative values in the +Lon selection, depleting positive selection coefficients and enriching for negative or null coefficients (*Figure 2A*). The number of advantageous mutations after reintroducing Lon decreased from 737 in –Lon selection to 384 in +Lon selection (*Figure 2B*), the mean selection coefficient for advantageous mutations decreased from 0.47 to 0.37, and the rank of the wild-type sequence increased by 341 to 864[th] (where 479 of the 863 higher-ranked variants are in the WT-like interval) (*Figure 2—figure supplement 3*). The median rank of the wild-type residue over all positions decreased from eight in –Lon selection to five in +Lon selection (*Figure 2—figure supplement 4*).

To examine in more detail how the mutational response of individual residues changes between selection ±Lon, we used a K-means clustering algorithm (see Materials and methods) to group all DHFR sequence positions into five categories: positions where mutations were generally advantageous (Beneficial), generally WT-like (Tolerant), variably advantageous and disadvantageous (Mixed), generally disadvantageous (Restricted), and generally null (Intolerant). Grouping was performed separately for –Lon and +Lon selection (*Figure 3—source data 1*). Comparing the distributions of DHFR positions in –Lon and +Lon conditions illustrates the extensive reshaping of the mutational landscape by Lon (*Figure 2C,D*). For –Lon selection, 28 positions (17.6%) were classified as Beneficial, where nearly every mutation was preferred over the wild-type residue. In comparison, the number of Beneficial positions decreased to 10 in +Lon selection, with only three surface-exposed positions (E48, T68, D127) common between the two Beneficial sets. Simultaneously, the number of Restricted positions increased from 42 to 67 with the reintroduction of Lon into the selection strain (*Figure 2C*). These results support the conclusion that Lon activity broadly penalizes mutations, including a large subset of the advantageous mutations. Overall, the changes upon modulating Lon activity lead to a model in which upregulating Lon increases constraints on DHFR, and the mutational landscape changes from being permissive when Lon is absent to being more restricted when Lon is present (*Figure 2D*).

To analyze the constraints imposed by Lon on the DHFR mutational landscape in structural detail, we defined a Δselection coefficient for each amino acid residue at each position as the difference between the +Lon and –Lon selections (*Figure 3A*). The Δselection coefficient values were most negative at positions in the Beneficial category and at positions with a native VILMWF or Y amino acid residue (*Figure 3B*, excludes Intolerant positions from –Lon selection); overall, mutations at positions

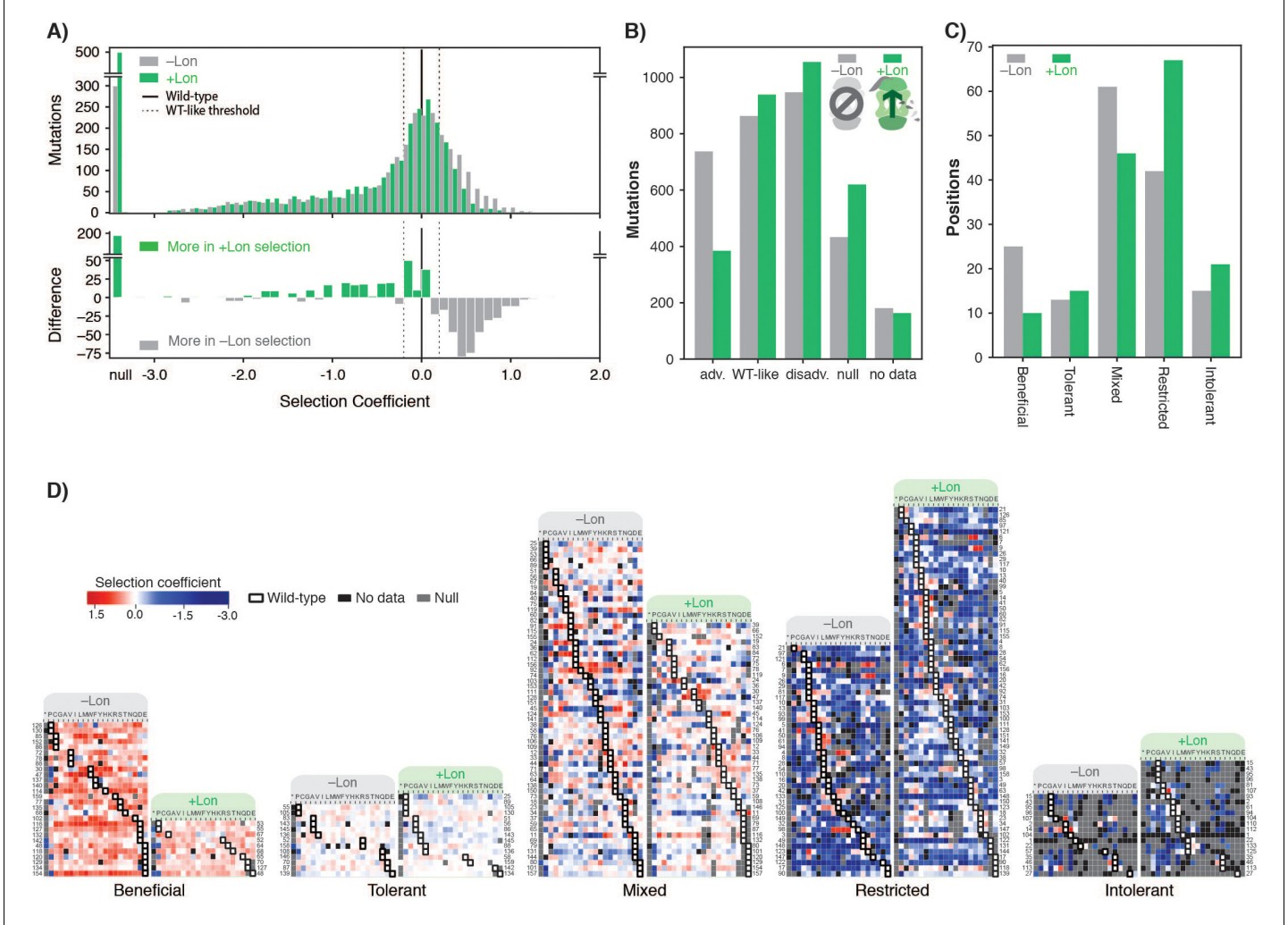

**Figure 2.** Lon protease expression reshapes the mutational landscape. (**A**) Histogram of selection coefficients for mutations (top) in –Lon (grey) and +Lon selection (green). The difference of the histograms (bottom) is shown with grey indicating more mutants for –Lon selection and green indicating more mutants for +Lon selection. The threshold for classification for advantageous and disadvantageous mutations is as in *Figure 1* and indicated with dashed lines. (**B**) Distribution of mutations classified by selection coefficients: $0.2 \leq$ advantageous (adv.), $0.2 >$ WT like $> -0.2$, $-0.2 \geq$ disadvantageous (disadv.), null, and no data (a mutant was not detected in the library after transformation into the selection strain). Grey bars: –Lon selection; green bars: +Lon selection. (**C**) Distribution of sequence positions into the five mutational response categories: Beneficial, Tolerant, Mixed, Deleterious, Intolerant. Grey bars: –Lon selection; green bars: +Lon selection. (**D**) Heatmap of DHFR selection coefficients in the –Lon and +Lon strains, showing details of the distributions shown in C) (dotted border). Positions (rows) are grouped by their mutational response category for –Lon and +Lon as in C) and sorted by the wild-type amino acid. Amino acid residues (columns) are organized by physiochemical similarity and indicated by their one-letter amino acid code. An asterisk indicates a stop codon. Advantageous mutations are shown in shades of red, disadvantageous mutations in shades of blue, Null mutations in grey and 'No data' as defined in A) in black. Wild-type amino acid residues are outlined in black.

The online version of this article includes the following figure supplement(s) for figure 2:

**Figure supplement 1.** Quality of the selection under +Lon conditions.

**Figure supplement 2.** Relationship between error and selection coefficient for +Lon selection.

**Figure supplement 3.** Comparison of selection coefficients ±Lon Scatterplot comparing selection coefficients in –Lon and +Lon selection, showing that mutations are generally repressed by Lon activity.

**Figure supplement 4.** Ranks of the wild-type amino acid residues in ±Lon selections.

**Figure supplement 5.** Comparison of DHFR per-position sequence preferences.

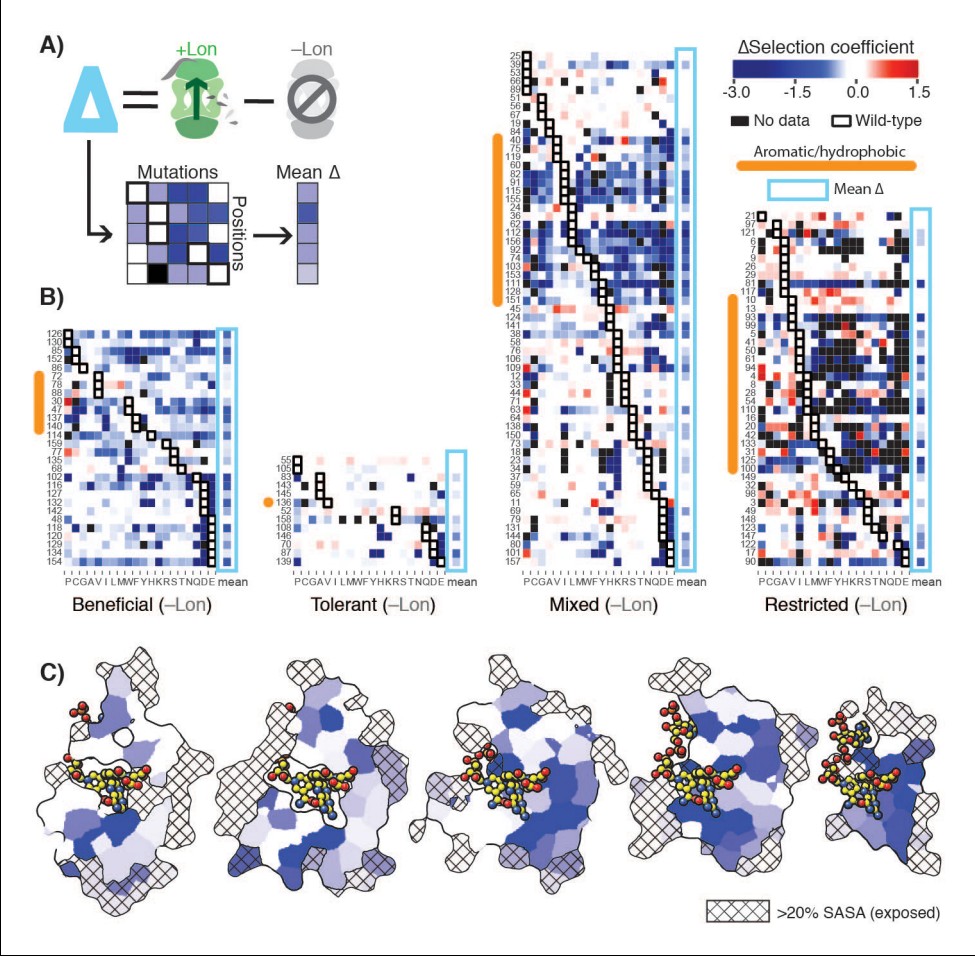

**Figure 3.** Delta selection coefficients show Lon impact. (**A**) Conceptual diagram of Δselection coefficients, calculated as the +Lon selection coefficient minus the −Lon selection coefficient (see Materials and methods). (**B**) Heatmap of Δselection coefficient values for all positions not classified as Intolerant. Δselection coefficients values between −0.2 and 0.2 are shown in white; Δselection coefficients >0.2 are in shades of red and Δselection coefficients <−0.2 in shades of blue. Amino acid residues (columns) are organized by physiochemical similarity and indicated by their one-letter amino acid code. The mean Δselection coefficient (avg) at each position is shown as a separate column and outlined with a light blue box. Positions (rows) are sorted by the wild-type amino acid and grouped by their mutational response category from the −Lon selection in *Figure 2C,D*. Positions with a native VILMWF or Y amino acid are indicated with an orange bar to the left. (**C**) Per-position mean Δselection coefficient displayed on the structural model of DHFR. The five cross-section slices of the DHFR structure are displayed as in *Figure 1E*, and the color scale is as in B).

The online version of this article includes the following source data and figure supplement(s) for figure 3:

**Source data 1.** Burial classification for DHFR positions from the Getarea server (*Fraczkiewicz and Braun, 1998*) as described in Materials and methods.

**Figure supplement 1.** Δselection coefficients.

with native hydrophobic residues are enriched for negative Δselection coefficients (*Figure 3—figure supplement 1A*). Strikingly, the mean Δselection coefficient was −0.71 for the 65 buried positions with <20% side-chain solvent accessible surface area, compared to −0.27 for the 79 exposed positions (*Figure 3C*, *Figure 3—figure supplement 1B*, *Figure 3—source data 1*). These results show that Lon has a broad impact on the mutational landscape throughout the DHFR structure but imposes particularly strong constraints in the DHFR core.

To determine why mutations in DHFR were advantageous in the absence of Lon but less so in its presence, we selected a subset of mutations for more detailed characterization in individual experiments. We considered all positions with more than one mutation in the top 100 most advantageous

mutations for the –Lon condition. We describe these positions by their location in one of four structural regions that appear to be hot-spots for the top advantageous mutations (*Figure 4A,B*, *Figure 4—figure supplement 1*): 1) exchanges between hydrophobic residues at core positions, 2) disruptions of surface residues on the beta-sheet near the active site, 3) disruptions of polar interactions with the adenine ring of NADPH, or 4) mutations to the active site or M20 loop that controls access to the active site. At these positions, we selected strongly advantageous mutations. Where possible, we selected two mutations at the same position but with significantly differing Lon sensitivities such that the set had a range of Δselection coefficients from −0.07 to −1.46, with the exception of L24V that had a positive Δselection coefficient. We first confirmed that the selected advantageous mutations indeed had higher cytosolic DHFR activity (the total rate of conversion of DHF to THF) in ER2566 Δ*folA*/Δ*thyA* (–Lon) lysates relative to the activity for WT DHFR (*Figure 4—figure supplement 2*), consistent with the deep mutational scanning results.

The lysate activity assay reports on both the enzymatic activity of a DHFR variant and its intracellular abundance, [DHFR] (*Bershtein et al., 2015b*; *Dykhuizen et al., 1987*). To separate the two contributions, we purified each of the DHFR variants and determined their enzymatic velocity in vitro using concentrations of DHF that are consistent with estimates of cytosolic DHF concentration based on mass spectrometry measurements (*Kwon et al., 2008*). At 20 μM DHF, 16 the mutants had velocities equal and up to three-fold higher than that of WT (*Figure 4C*, *Figure 4—figure supplement 3*, *Figure 4—source data 1*). In contrast, the other eight mutants had velocities as much as two-fold lower than that of WT at the same DHF concentration. These results show that the higher cytosolic DHFR activity of the advantageous mutations can only partially be explained by changes in the kinetic parameters for these mutants.

We therefore examined the soluble intracellular abundance of these mutants. In the absence of Lon, we observed that mutant abundance levels varied from close-to-wild-type levels to a 20-fold increase over wild-type (*Figure 4D*, *Figure 4—figure supplement 4*, *Figure 4—source data 2*). Importantly, abundance decreased for most mutants in the presence of Lon (*Figure 4—figure supplement 4*), as expected, and these abundance decreases correspond to decreased selection coefficients (negative values in the Δselection coefficients from *Figure 3* that report on the Lon impact on selection (*Figure 4—figure supplement 5*)). Moreover, when considering both velocity and abundance the expected total cellular DHFR activity ([DHFR] • velocity) is increased compared to wild-type for the majority of advantageous mutants (*Figure 4E*, *Figure 4—figure supplement 6*, positions above the dotted line indicate expected cellular activity greater than wild-type). However, the expected total cellular DHFR activity is not a strong quantitative predictor of the advantageous mutants in –Lon selection (*Figure 4—figure supplement 7*, *Figure 4—figure supplement 8*). We attribute discrepancies at least in part to the difficulty of accurately quantifying rather small differences in activity and abundance, in addition to other potential complicating factors such as differential activity of cellular chaperones for different DHFR variants (*Cho et al., 2015*), and feedback regulation that could affect cellular concentrations of the substrate DHF (*Bershtein et al., 2015a*; *Kwon et al., 2008*). Nevertheless, our velocity and abundance measurement are in qualitative agreement with the in vivo selection. Taken together, these results suggest that increased selection coefficients arise from an interplay of effects of the mutations on cellular abundance and catalytic activity (*Dykhuizen et al., 1987*), and that each parameter alone is insufficient to explain the majority of the advantageous mutations. Moreover, Lon suppresses advantageous mutations at least in part by reducing their cellular abundance.

To test more directly whether advantageous mutations in DHFR destabilize the protein and whether this destabilization could explain the sensitivity to Lon expression, we measured apparent melting temperature ($T_m$) values from non-reversible thermal denaturation monitored by circular dichroism spectroscopy. We found that many of the advantageous mutations considerably destabilized the protein (*Figure 4F*, *Figure 4—figure supplement 9*, *Figure 4—source data 3*). Moreover and as expected, the Δselection coefficients between +Lon and –Lon selection (*Figure 3*) are correlated with $T_m$ (*Figure 4F*), except for mutations near the active site. Strikingly, when we compare different mutations at the same position, the change in Δselection coefficients (i.e. Lon sensitivity) correlates with the change in $T_m$ values (*Figure 4G*). These results indicate that the many of the selected advantageous mutations are destabilizing, and that destabilization is correlated with Lon sensitivity. One possible explanation for the selection advantage of the subset of destabilizing mutations with increased $k_{cat}$ (e.g. L24V, W30F/M, M42F/Y, H114V, D116I/M, E154V) is that these

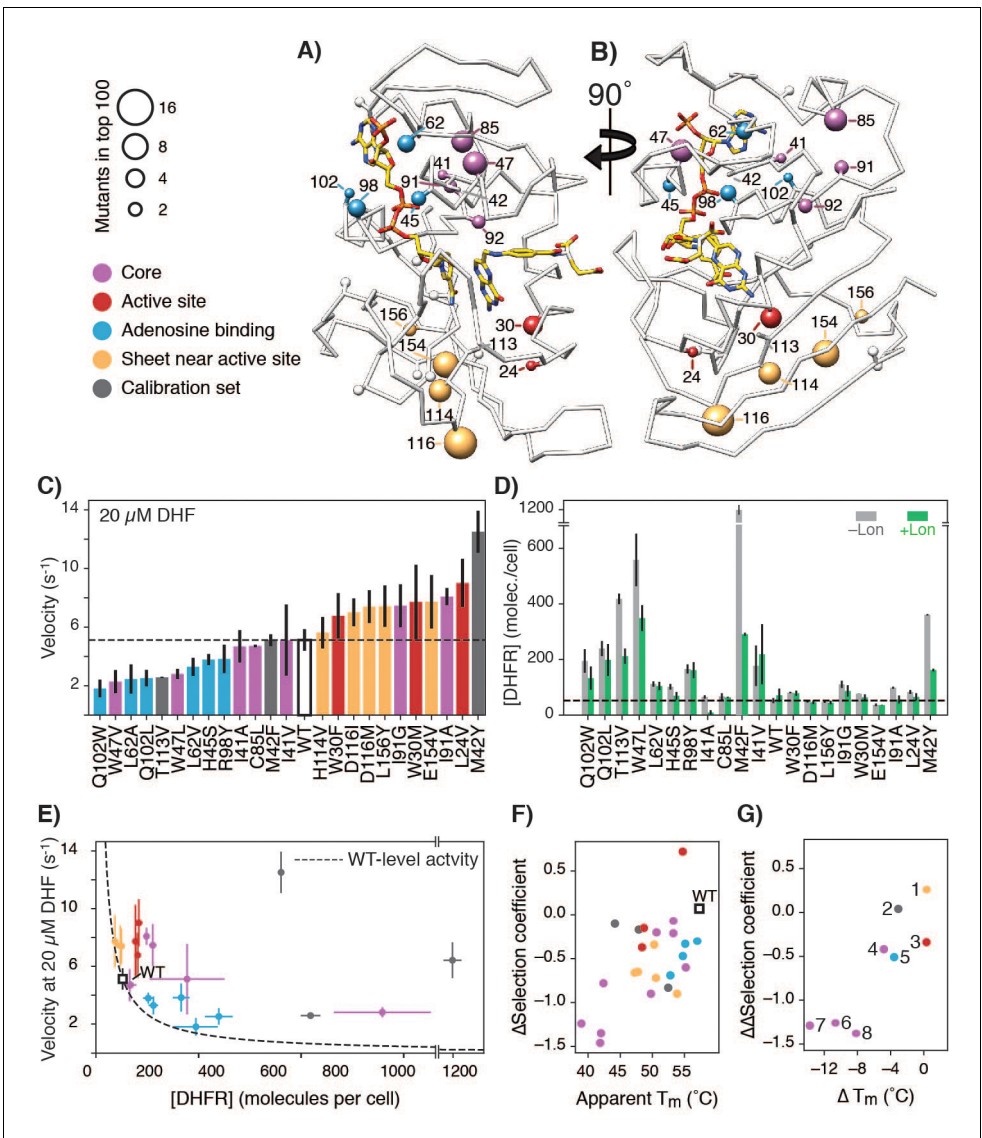

**Figure 4.** Advantageous mutations arise from an interplay of increased enzymatic velocity and increased abundance in the absence of Lon. (**A**) DHFR structure with mutational hot-spots. For positions with two or more top 100 advantageous mutations in the absence of Lon, the beta carbon is depicted as a sphere scaled according to the number of top mutations. For mutants selected for in vitro characterization, the beta carbon is colored according to its location in the DHFR structure: core (purple), surface beta-sheet (gold), proximal to the adenine ring on NADPH (blue), or proximal to the active site and M20 loop (red). Positions for advantageous mutants from the calibration set are depicted in dark grey. (**B**) The structure from A) rotated 90° clockwise. (**C**) In vitro velocities of purified DHFR wild-type and point mutants measured at 20 µM DHF. Bars are colored in reference to the hot-spots in A). Error bars represent ±1 standard deviation from three independent experiments (Materials and methods). The dashed line represents the velocity of WT DHFR. (**D**) DHFR cellular abundance calculated from the lysate DHFR activity in *Figure 4—figure supplement 2* and in vitro kinetics with purified enzyme (see Materials and methods). Error bars represent the cumulative percent error (standard deviation) from three independent experiments for velocity and three biological replicates for lysate activity. Data are shown in both the -Lon (light grey) and +Lon (green) conditions. The dashed line represents the WT expression level of DHFR in the –Lon background. Mutants are in the same order as in C) (see *Figure 4—source data 2*; four mutants were not measured). (**E**) Cellular abundance of DHFR vs. in vitro velocities of purified DHFR wild-type and point mutants measured at 20 µM DHF. Points are colored as in A). Error bars represent ±1 standard deviation from three independent experiments (Materials and methods). The dashed line represents WT-level DHFR activity, i.e. DHFR abundance/velocity pairs whose product is equivalent to [DHFR]$_{WT}$ • velocity$_{WT}$. (**F**) Correlation between in vitro T$_m$ values and in vivo Δselection coefficients for DHFR wild-type and characterized mutants. Points are

*Figure 4 continued on next page*

*Figure 4 continued*

colored as in A). (G) $\Delta T_m$ values and $\Delta\Delta$selection coefficient for mutations at the same position. Points representing comparison between mutants are numbered as follows: 1) D116I-M, 2) M42Y-F, 3) W30M-F, 4) I91G-A, 5) Q102W-L, 6) L62A-V, 7) I41A-V, 8) W47V-L.

The online version of this article includes the following source data and figure supplement(s) for figure 4:

**Source data 1.** In vitro velocity for selected advantageous measured as described in Materials and methods at multiple concentrations of DHF are reported with the standard deviation over three independent experiments.

**Source data 2.** Soluble DHFR abundance levels in molecules per cell measured from lysate activity assays as described in Materials and methods.

**Source data 3.** Apparent $T_m$ values from thermal denaturation experiments monitored by CD signal at 225 nm are reported along with the $\Delta$selection coefficient (Lon impact) value depicted in *Figure 4D*.

**Figure supplement 1.** Structural context for hotspot residues from *Figure 4*.

**Figure supplement 2.** Lysate activity for DHFR wild-type and point mutants on the selection plasmid.

**Figure supplement 3.** In vitro velocities of purified DHFR wild-type and point mutants.

**Figure supplement 4.** Soluble cellular abundance for DHFR wild-type and point mutants on the selection plasmid.

**Figure supplement 5.** Lon impact as $\Delta$selection coefficient versus change in DHFR abundance ±Lon.

**Figure supplement 6.** Cellular abundance versus in vitro velocity for DHFR wild-type and point mutants.

**Figure supplement 7.** Selection coefficient compared to predictions of DHFR wild-type and point mutant activity from cellular abundance and in vitro velocity measurements.

**Figure supplement 8.** Zoom in for Selection coefficient compared to predictions of DHFR wild-type and point mutant activity from cellular abundance and in vitro velocity measurements.

**Figure supplement 9.** Thermal denaturation curves monitored by CD signal at 225 m for selected hotspot mutants.

---

mutations promote breathing motions that accelerate product release, which is rate limiting for wild-type DHFR at neutral pH (*Oyen et al., 2017*) and for a hyperactive DHFR mutant with a 7-fold increase in $k_{cat}$ (*Iwakura et al., 2006*).

Taken together, our data indicate that the observed widespread changes in the mutational landscape of DHFR can be explained by a penalty for destabilizing mutations from Lon expression, leading to extensive activity – stability tradeoffs for advantageous mutations. The effect of these two selection pressures is directly observable in the structural arrangement of the mutational response categories (*Figure 5*, *Figure 5—figure supplement 1*). In –Lon conditions, mutational responses are arranged in shells around the hydride transfer site (*Liu et al., 2013*; *Figure 5A*, top), where the proportion of advantageous mutations increases with increasing distance (*Figure 5B*). This same spatial pattern also holds for +Lon selection (*Figure 5A*, bottom), but it is now superimposed with the additional pressure against destabilizing mutations such that there are no Beneficial positions in the core (*Figure 5C*, *Figure 5—figure supplement 2*). In contrast, the mutational responses as a function of distance to other DHFR sites (e.g. C5 of the NADPH adenine ring) do not show as strong of a relationship (*Figure 5—figure supplement 3*). These findings illustrate how the contributions from two constraints – one structural (distance from hydride transfer) and one dependent on cellular context (Lon) – can be distinguished from structural patterns in the mutational landscape.

## Discussion

The naturally occurring insertion in the Lon promoter in our original selection strain, in combination with our stringent selection conditions, allowed the serendipitous discovery that advantageous mutations are remarkably prevalent throughout the DHFR structure but are also highly sensitive to Lon. The large fraction of advantageous mutations to DHFR appears to conflict with the fixation of the wild-type DHFR sequence during evolution. While Lon expression in our selection increases both the relative rank of the WT DHFR sequence (*Figure 2—figure supplement 4*) and the similarity between amino acid preferences from selection and from bacterial DHFR orthologues (*Figure 2—figure supplement 5*), there are still considerable differences: There are still 384 advantageous mutants that rank substantially better than the WT sequence even in the presence of Lon, and the amino preferences in the two selection experiments (±Lon) are more similar to each other than either is to the preferences from bacterial DHFR orthologues.

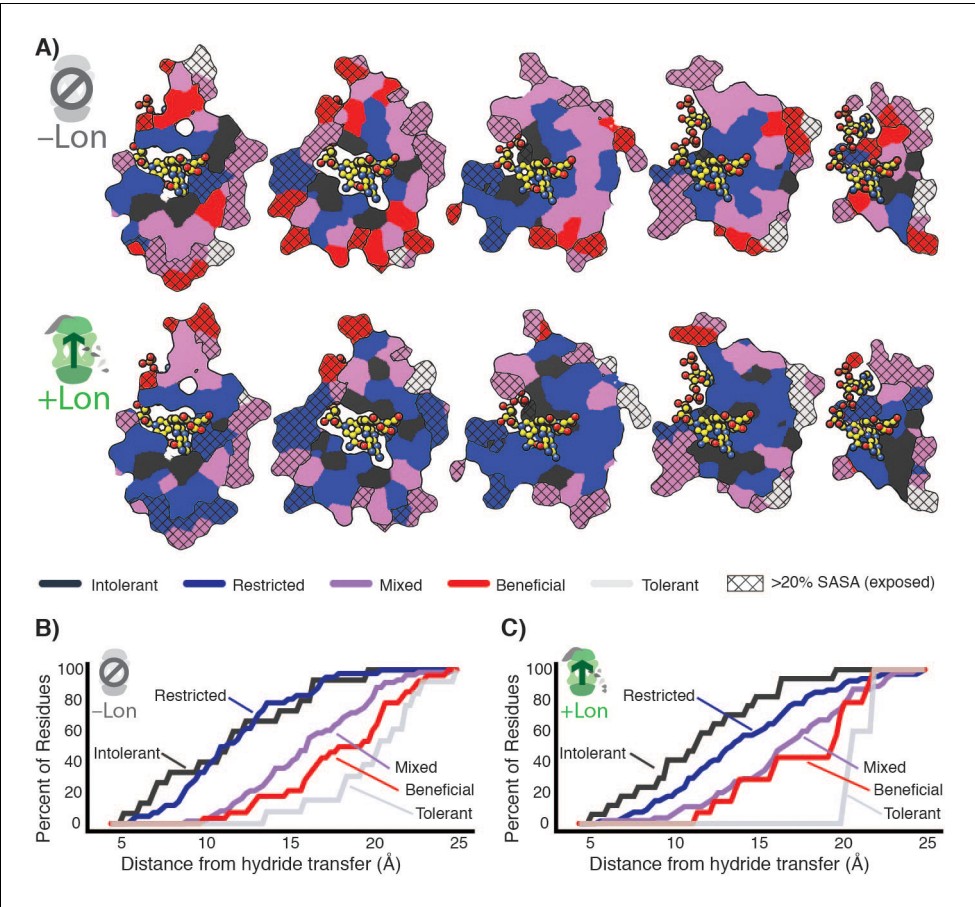

**Figure 5.** Structural characterization of multiple constraints on the DHFR mutational landscape. (**A**) Mutational response categories from −Lon selection (top, categories in *Figure 2C,D*) and +Lon selection (bottom, categories as in *Figure 2C,D*) colored onto residues and displayed on slices as in *Figure 1E*. (**B**) Relationship between mutational response and distance from hydride transfer for −Lon selection. The percent of positions from each mutational response category are plotted as a function of distance from the site of hydride transfer. Each category colored as in **A**), top). (**C**) Relationship between mutational response and distance from hydride transfer for +Lon selection. Each category colored as in **A**), bottom).

The online version of this article includes the following figure supplement(s) for figure 5:

**Figure supplement 1.** Selection coefficients under the two Lon expression regimes mapped on the DHFR structure.

**Figure supplement 2.** Burial of residues within each mutation response category reported as the mean number of atomic neighbors.

**Figure supplement 3.** Residues in mutational response categories in the −Lon selection as a function of distance from several sites in the DHFR structure.

Considering these differences, we note several caveats in comparing our selection results to selection in evolution: First and most generally, screening DHFR variants under calibrated selection conditions (such as defined temperature, medium, and growth kept in early log phase) for a few generations is not expected to recapitulate the natural selection pressures on *E. coli* DHFR on evolutionary timescales. Second and more specifically, our selection conditions were intentionally engineered to be highly sensitive to mutations by dampening DHFR abundance to approximately 10% of the endogenous level (*Figure 1—figure supplement 2*). In contrast, endogenous DHFR is expected to be buffered from mutational impacts. Increasing DHFR activity or abundance in *E. coli* several-fold above that in wildtype strains does not increase fitness, and, conversely, reducing DHFR abundance in *E. coli* does not have an impact on growth until abundance is below 30% of the endogenous

level (*Bershtein et al., 2013*; *Bhattacharyya et al., 2016*). Indeed, selection on mutations to the chromosomal DHFR gene did not reveal strong mutational impacts in the absence of the anti-folate drug trimethoprim (*Garst et al., 2017*). Third, chromosomal DHFR expression is modulated through feedback mechanism (*Bershtein et al., 2015a*), and it would be an interesting question how the distribution of fitness effects of DHFR mutations will be shaped by the presence of such a regulatory expression element that is absent in our selection system. Taken together, these mutational buffering effects likely explain why mutations that are advantageous in our selection are not prevalent in evolutionary DHFR sequences, and likely also explain why DHFR sequences do not vary between naturally occurring –Lon and +Lon strains of *E. coli*.

Nevertheless, our engineered selection conditions yielded considerable insights into constraints on mutational landscapes that are typically hidden from observation precisely because of buffering effects in natural contexts. The increase in the number of advantageous mutations in the absence of Lon shows that decreasing cellular constraints can substantially modulate the tolerance to mutation in a deep mutational scanning experiment. Because all B type *E. coli* strains (e.g. BL21) have the same natural Lon deficiency as our selection strain, our results could have implications for selection experiments performed in these strains over much longer time-scales such as the *E. coli* Long-Term Evolution Experiment (*Tenaillon et al., 2016*), or directed evolution strategies that often lead to mutations at positions distal to the active site.

Beyond experiments in B-type *E. coli*, we expect the fundamental principle of tuning trade-offs to play a role in other experimental systems. Prior work has illuminated the impact of chaperones on the effect of mutations, such as for GroEL in bacteria (*Tokuriki and Tawfik, 2009*) and for Hsp90 in eukaryotic cells that has been shown to buffer the phenotypic impacts of deleterious mutations (*Queitsch et al., 2002*). Our results highlight an opposite key role for the protein quality control machinery to tune in vivo mutational responses and lead to a model where protease activities add constraints to the mutational landscape and chaperones relieve them.

The ability to tune multiple constraints could provide a general way of controlling landscapes to drive genes into regions of sequence space that are highly responsive to external pressures. A concrete example of how this principle could be applied is in combinatorial antibiotics. Lon inactivation has been shown to increase resistance to antibiotics (*Nicoloff and Andersson, 2013*). Switching between compounds capable of inhibiting or activating Lon in combination with DHFR-targeting folate inhibitors such as trimethoprim could serve to variably promote destabilized resistance mutants when Lon is inhibited and then penalize those mutations when Lon is reactivated.

While the power in engineering individual gene sequences is well-recognized, we are only just beginning to explore the potential in engineering the general behavior of local sequence space. We anticipate that further study of tunable constraints will yield a new toolkit for fine control of the landscapes that guide movements through sequence space and enable unexplored engineering applications.

## Materials and methods

All plasmid and primer sequences are listed in The Appendix. Key plasmids were deposited in the Addgene plasmid repository (accession codes are listed in The Appendix). All code and python scripts are available at https://github.com/keleayon/2019_DHFR_Lon.git with key input files and example command lines (*Thompson, 2020*; copy archived at https://github.com/elifesciences-publications/2019_DHFR_Lon).

### Generation of plasmids for in vivo selection assay

The vector bearing DHFR and TYMS for in vivo selection (SMT205) was derived from the pACYC-Duet vector described by *Reynolds et al., 2011*. The lac operon upstream of the TYMS gene was replaced with a Tet-inducible promoter. A Tet promoter fragment had been generated with overlap extension PCR and cloned into the pACYC vector (SMT101) at unique AflII/BglII sites to produce SMT201. Selection conditions that resolved increased-fitness mutations were obtained with the SMT205 plasmid where the DHFR 'AAGGAG' ribosome binding site (RBS) was replaced with 'AATGAG' based on prediction from the RBS calculator (*Salis et al., 2009*) using inverse PCR. Briefly, PCR reactions were set up using 2x Q5 mastermix (NEB, cat# M0492), 10 ng of plasmid template, and 500 nM forward and reverse primers. PCR was performed in the following steps: 1) 98 °C for 30

s, 2) 98 °C for 10 s, 3) 57–63 °C for 30 s, 4) 72 °C for 2 min, 5) return to step 2 for 22 cycles, 6) 72 °C for 5 min. As needed, the annealing temperature (step 3) was optimized in the range of 57–63 °C. 25 µL of PCR reaction was mixed with 1 µL of DpnI (NEB, cat# R0176), 1 µL of T4 PNK (NEB, cat# M0201), 1 µL of T4 ligase (NEB, cat# M0202), and 3.1 µL of T4 ligase buffer (NEB, cat# B0202) at 37 °C for 2–4 hr. The reactions were then transformed into chemically competent Top10 cells and plated on LB agar plates with 35 µg/mL chloramphenicol (Fisher BioReagents, BP904, CAS: 56-76-7, 35 mg/mL in ethanol). The plates were incubated overnight at 37 °C. Single colonies were picked and used to inoculate 5 mL of LB medium (10 g Bacto-tryptone (Fisher BioReagent, cat# BP1415, CAS: 73049-73-7), 5 g Bacto-yeast extract (BD Difco, cat# 212720, CAS: 8013-01-2), 10 g NaCl (Fisher BioReagents, cat# BP358, CAS 7647-14-5), 0.186 g KCl (Sigma, cat# P9541, CAS: 7447-40-7), volume brought to 1 L with MilliQ water, autoclaved) + 35 mg/mL chloramphenicol. Cultures were incubated overnight in 14 mL plastic culture tubes (Falcon, cat# 352059) at 37 °C under 225 rpm shaking. Pellets were collected by centrifugation at 3500 rpm for 10 min at 4 °C in a swinging-bucket centrifuge (Beckman Coulter, Allegra X-12R) and miniprepped (Qiagen, cat# 27104). Constructs were confirmed by Sanger sequencing (Quintara Biosciences) by alignment to the template sequence in ClustalOmega.

## Generation of plasmid libraries

Four sublibraries were generated to cover the entire mutational space of *E. coli* DHFR: positions 1–40 (sublibrary1, SL1), positions 41–80 (sublibrary2, SL2), positions 81–120 (sublibrary3, SL3), and positions 121–159 (sublibrary4, SL4). The single point mutant library was performed by multiple parallel inverse PCR reactions to substitute an NNS degenerate codon at every codon in DHFR. PCR primers (The Appendix) were phosphorylated in a 20 µL reaction with 1 µL T4 polynucleotide kinase and 1x T4 ligase buffer. Inverse PCR reactions were performed as described above, followed by PCR clean-up (Qiagen, cat# 28104). The cleaned PCR reactions were incubated for 4 hr with 1 µL DpnI, 1 µL of T4 ligase, and 3 µL of T4 ligase buffer. PCR reactions were analyzed by gel electrophoresis using a 1% agarose gel in TAE buffer (20 mM acetic acid (Sigma Aldrich, cat#, 695092, CAS: 64-19-7), 2 mM EDTA (ACROS Organics, cat# AC118432500, CAS: 60-00-4), 40 mM Tris, pH 8.5) with 0.01% v/v GelRed (Biotium, cat# 41003), and the product amount was quantified using gel densitometry in the FIJI image processing software package (*Schindelin et al., 2012*). Samples were pooled stoichiometrically, cleaned once with a gel extraction kit (Qiagen, cat# 28115), and again with a PCR clean-up kit. The pooled and cleaned ligation products were transformed into *E. coli* Top10 cells by electroporation (BioRad GenePulser Xcell, 1 mm path length cuvette (cat# 165–2089), 1.8 kV, time constant ~5 ms) using ~5 µL to obtain a minimum of $10^7$ transformants as measured by dilution plating on LB-agar plates with 35 µg/mL chloramphenicol. The transformed cells were rescued in SOB medium (20 g Bacto-tryptone, 5 g Bacto-yeast extract, 0.584 g NaCl, 0.186 g KCl, 800 mL MilliQ water, pH 7.0, volume brought to 1 L with MilliQ water, autoclaved) without antibiotics for 45 min at 37 °C before culturing overnight in 10 mL SOB medium with 35 µg/mL chloramphenicol. In the morning, glycerol stocks were made by mixing 500 µL of saturated culture with 500 µL of sterile filtered 50% (v/v) glycerol. 5 mL of the culture was used to miniprep the transformed library with a Qiagen miniprep kit.

## Generation of individual point mutant plasmids

Point mutants in all DHFR-containing plasmids were generated via inverse PCR as described above for the generation of SMT205 except that the appropriate antibiotic was matched with the plasmid (The Appendix). Library primer sequences (The Appendix) were used except that the 'NNS' sequence on the forward primer was replaced with the desired codon.

## Generation of ER2566 ΔfolA ΔthyA –Lon and ER2566 ΔfolA ΔthyA +Lon

The *ER2566 ΔfolA ΔthyA –Lon* strain was generated as previously described (*Reynolds et al., 2011*) and a gift from Prof. Stephen Benkovic. The ER2566 *ΔfolA ΔthyA +Lon* strain was generated from ER2566 *ΔfolA ΔthyA –Lon* by lambda red recombination using Support Protocol I from *Thomason et al., 2014*. The pSim6 plasmid bearing the Lamda red genes linked to a temperature sensitive promoter and the pIB279 plasmid bearing the Kan-SacB positive-negative selection

marker (*Blomfield et al., 1991*) were gifts from Carol Gross. The Kan-SacB cassette was amplified with 2 rounds of PCR using primers with 5' homology arms for the region upstream of the Lon gene (The Appendix). The insertion fragment containing the Anderson consensus promoter (*iGEM, 2006*) with homology arms for the region upstream of Lon in the ER2566 genome was amplified from primers using overlap extension PCR.

## Plate reader assay for *E. coli* growth

Growth rates for the selection strains bearing individual DHFR mutants were measured in 96-well plate growth assays as described for one individual mutant. The SMT205 plasmid was transformed via heat shock into chemically competent *ER2566 ΔfolA ΔthyA ±Lon* cells and plated on an LB-agar plate with 30 μg/mL chloramphenicol plus 50 μg/mL thymidine and incubated overnight at 37 ˚C. On the second day, 2 mL M9 medium (1x M9 salts (BD Difco, cat# 248510), 0.4% glucose w/v (Fisher Chemical, cat# D16, CAS: 50-99-7), 2 mM MgSO4 (Sigma Aldrich, cat# 63138, CAS:10034-99-8)) with supplements for deficient folate metabolism (50 μg/mL thymidine (Sigma Aldrich, cat# T1895, CAS: 50-89-5), 22 μg/mL adenosine (Sigma Aldrich, cat# A9251, CAS: 56-61-7), 1 μg/mL calcium pantothenate (TCI, cat# P0012, CAS: 137-08-6), 38 μg/mL glycine (Fisher BioReagents, cat# BP381, CAS: 56-40-6), and 37.25 μg/mL methionine (Fisher BioReagents, cat# BP388, CAS 63-68-3)) and 30 μg/mL chloramphenicol in a 14 ml culture tube was inoculated with 5–10 colonies scraped from the plate and incubated at 37 ˚C at 225 rpm shaking for 12–14 hr. Biological replicates were obtained from separate inoculations at this step and run on the same plate. All assays were run from fresh transformations. Then, 20–50 μL of the previous culture was used to inoculate 5 mL of M9 medium (no supplements) with 30 μg/mL chloramphenicol in a 14 ml culture tube. This fresh culture was incubated for 6 hr at 30 ˚C at 225 rpm shaking. Meanwhile 2 mL of M9 medium with 30 μg/mL chloramphenicol and a transparent 96-well plate were pre-warmed at 30 ˚C. After the 6 hr incubation, the optical density at 600 nm (OD600) of the culture was measured on a Cary 50 spectrophotometer over a path of 1 cm. This early log-phase culture was diluted to an OD600 = 0.005 in the 2 mL aliquot of warmed M9. 200 μL of the dilute culture was pipetted into a well in the 96-well plate. Technical replicates were obtained by dispensing the same dilute culture into multiple wells. Wells were covered with 50 μL of mineral oil (Sigma Aldrich, cat# M5904, CAS: 8042-47-5) using the reverse pipetting technique. The plate was then incubated for 20–48 hr at 30 ˚C in a Victor X3 multimode plate reader (Perkin Elmer). Every 10 min, the plate was shaken for 30 s with an orbital diameter of 1.8 mm under the 'normal' speed setting. Then, the absorbance at 600 nm (ABS600) was measured for each well. Growth rates were calculated from the slope of $Log2(ABS600 - ABS600_{t=0})$ for $\Delta ABS600$ in the range of 0.015–0.04 using an in-house python script.

## Deep mutational scanning experiments

Competitive growth under selection for cellular DHFR activity was performed in a continuous culture turbidostat (gift of Rama Ranganathan) as described below for a single sublibrary. Sublibraries of DHFR single point mutants were transformed via electroporation as described above into electrocompetent *ER2566 ΔfolA ΔthyA ±Lon* cells using approximately 50 ng of plasmid DNA and 80 μL of competent cells with a transformation efficiency of $10^8$ cfu/ng (based on testing with 10 ng of pACYC plasmid DNA). Immediately after electroporation, the cells were rescued with 2 mL of SOB medium with 50 μg/mL thymidine warmed to 37 ˚C. The rescue culture was incubated at 37 ˚C for 45 min at 225 rpm shaking. After the rescue step, 4 μL of the rescue medium (1/500 of the rescue volume) was serially diluted in 10-fold increments. Half the volume of each dilution (1/1000 – $1/10^7$ of the rescue volume) was plated on an LB-agar plate with 30 μg/mL chloramphenicol plus 50 μg/mL thymidine and incubated overnight at 37 ˚C. The colonies were counted the following morning to check for a minimum of 1000x oversampling of the theoretical diversity in the library ($\sim 10^6$ transformants for each sublibrary). Meanwhile, the larger portion of the rescue medium was mixed with 4 mL of SOB medium with 45 μg/mL chloramphenicol (1.5x) plus 50 μg/mL thymidine warmed to 37 ˚C. This 6 mL culture was incubated for 5–6 hr at 37 ˚C at 225 rpm shaking in a 14 mL culture tube. After incubation, the culture was pelleted by centrifuging for 5 min at 3000 rpm at room temperature in a swinging bucket centrifuge. The cells were resuspended in 50 mL of supplemented M9 medium + 30 μg/mL chloramphenicol and incubated for 12–14 hr at 37 ˚C at 225 rpm shaking in a 250 mL flask. In the morning, 150 mL of supplemented M9 medium + 30 μg/mL chloramphenicol in a 1 L flask was

inoculated with 15 mL of the overnight culture. This pre-culture was incubated at 30 °C for 4 hr at 225 rpm shaking. After 4 hr, the pre-culture was centrifuged at 3000 rpm for 5 min at room temperature in a swinging bucket centrifuge, and the OD600 was measured to ensure that the culture did not grow beyond early-mid log phase (OD600 ~0.3). The supernatant was decanted, and the pellet was resuspended in 30 mL of M9 medium. Pelleting and resuspension were repeated for a total of 3 washes to remove the supplemented medium. After three washes, the OD600 was measured for the resuspended pellet using a 10-fold dilution to stay in the linear range of the spectrophotometer.

The washed pellet was then transferred to the growth chamber of the turbidostat (a 250 mL pyrex bottle) containing 150 mL of M9 medium with 50 µg/mL chloramphenicol. Selection experiments were performed with 2 of the four sublibraries at a time (two repeats of SL1-SL2 and SL3-SL4, and one repeat of SL1+SL3 and SL2+SL4 for a net of biological triplicates for every codon in the gene), and the resuspended pellet from each library was diluted in the initial culture to an OD600 = 0.035. Mixing and oxygenation was provided by sterile filtered air from an aquarium pump. Every 60 s, the aquarium pump was stopped, and the optical density of the culture was read by an infrared emitter-receiver pair. The ADC (analog-to-digital converter) of the voltage over the receiver was calibrated against a spectrophotometer to convert the signal into an approximate OD600. The cells were grown at 30 °C with an OD600 threshold of 0.075. When the OD600 of the selection culture exceeded the threshold, the selection culture was diluted to OD600 ~0.065 with 25 mL of M9 medium with 50 µg/mL chloramphenicol, and the additional culture volume was driven through a waste line by the positive pressure of the aquarium pump. At timepoints of t = 0, 2, 4, 6, 8, 12, 16, and 18 hr, 6 mL of the selection culture in 2 mL centrifuge tubes was pelleted at 5000 rpm for 5 min at 4 °C in a microcentrifuge (Eppendorf, 5242R). The supernatant was removed except for the last ~200 µL, and the tubes were again pelleted at 5000 rpm for 5 min at 4 °C in a microcentrifuge, and all the supernatant was carefully removed from the pellet. The pellets were stored at −20 °C until sequencing.

## Amplicon generation

Amplicons were generated by two rounds of PCR. The first round of PCR amplifies a portion of the DHFR gene from the pACYC plasmid containing 2–3 sublibraries. For quality control templates were 1 ng/µL plasmid solutions and the amplicons covered SL1-SL2 or SL3-SL4. Round 1 PCR reactions were set up using 1 µL of template, 1% v/v Q5 hotstart polymerase (NEB, cat# M0493), 1x Q5 Reaction Buffer, 1x Q5 High GC Enhancer, 200 µM dNTPs, and 500 nM forward and reverse primers. PCR was performed in the following steps: 1) 98 °C for 30 s, 2) 98 °C for 10 s, 3) 57 °C for 30 s, 4) 72 °C for 12 s, 5) return to step 2 for 16 cycles, 6) 72 °C for 2 min.

The Round 2 PCR uses primers that attach the Illumina adapters and the i5 (reverse) and i7 (forward) barcodes for sample identification and demultiplexing. Round 2 PCR reactions were set up and run identically to Round one reactions except that the template was 1 µL of Round 1 PCR. Round two reactions were analyzed by gel electrophoresis using a 1% TAE-agarose gel in TAE buffer with 0.01% v/v GelRed, and the product amount was quantified using gel densitometry in FIJI. Samples were pooled stoichiometrically and cleaned with a gel extraction kit (Qiagen). Because of the risk of contamination from small primer dimers, gel extraction was performed with very dilute samples. Only 20 µL of sample was loaded onto a 50 mL TAE-agarose gel (OWL EasyCast, B1A) with 8 of the 10 wells combined into a single well. The pooled amplicons were then cleaned again with a PCR clean-up kit (Zymogen, cat# D4013) to allow for small volume elution. The final amplicon concentration was measured with a NanoDrop One UV spectrophotometer and by Picogreen assay (Thermo Scientific, cat# P11496).

## Sequencing for deep mutational scanning experiments

Templates for amplicon PCR were prepared from the frozen pellets. The pellets were resuspended in 20 µL of autoclaved MilliQ water and incubated on ice for 10 min. The samples were then centrifuged at 15,000 rpm for 10 min at 4 °C in a benchtop microcentrifuge. 1 µL of the supernatant was used as template in the amplicon generation protocol for sublibraries described above. The amplicons were sequenced on an Illumina NextSeq using a 300-cycle 500/550 high-output kit. Because of the limitations in the number of sequencing cycles on the Illumina NextSeq, the full amplicon was not sequenced for amplicons containing non-adjacent sublibraries (SL1+SL3, and SL2+SL4). Reads

were demultiplexed into their respective selection experiment and timepoint using their TruSeq barcodes. Paired end reads were joined using FLASH (*Magoč and Salzberg, 2011*). For amplicons with adjacent sublibraries (SL1-SL2 and SL3-SL4), the joined reads were kept. For amplicons with distal sublibraries (SL1+SL3 and SL2+SL4), the unjoined reads were kept. Reads from all lanes of the Illumina chip were concatenated and raw counts of DHFR mutants were obtained from these reads.

Reads on the Illumina NextSeq (two-color chemistry, LED optics) generally have lower quality scores than reads from the Illumina MiSeq (four-color chemistry, laser optics). This lower quality leads to a background signal. This background was estimated from a WT sample. The median + one standard deviation value of background count was subtracted from every allele and the alleles were translated into the amino acid sequence, combining synonymous sequences. Counts at each timepoint were only reported for an allele if its frequency was above $2.0 \times 10^{-5}$. Raw counts are reported in *Supplementary file 3–5*.

## Analysis of deep mutational scanning data

Mutant counts were used to generate selection coefficients on our background-subtracted count files with Enrich2 using unweighted linear regression (*Rubin et al., 2017*). The raw Enrich2 values for each unique selection experiment were combined with a post-processing script. Enrich2 does not calculate selection coefficients for mutants that have no counts at a timepoint, so some selection coefficients were recalculated using only the timepoints before the counts for that allele fell below the cutoff frequency of $2.0 \times 10^{-5}$. Individual selection coefficients were evaluated based on two criteria: noise and number of timepoints. Individual selection coefficients were discarded 1) if the standard error from regression was greater than 0.5 + 0.5 • |selection coefficient| or 2) if there were fewer than four timepoints reporting on the mutant. The regression for the fitness value of the mutants from replicate selection experiments to the average values across all experiments was calculated and the fitness values in each replicate were scaled to correct for linear differences in the selection values between replicates. These normalized values were then averaged for the final fitness value. Averaged selection coefficients values were evaluated based on two criteria: the standard deviation of the averaged selection coefficients and the number of replicates. Averaged selection coefficients were discarded 1) if the standard deviation over the normalized replicates was greater than 0.5 + 0.25 • |selection coefficient| or 2) if there were fewer than two replicates. In *Supplementary file 1* and *2* the fitness is reported as the mean normalized fitness, the standard error is reported as the combined Enrich2 standard error (from linear regression of timepoints), and the standard deviation is reported as the standard deviation of the biological replicates. The correlation and R-values of normalized replicate experiments and the distribution of standard deviations and standard errors for each mutant are reported in *Figure 1—figure supplement 3*, *Figure 1—figure supplement 4*, *Figure 2—figure supplement 1*, *Figure 2—figure supplement 2*.

Selection was evaluated by comparing selection coefficients to DHFR velocity from reported Michaelis-Menten kinetics at cytosolic concentrations of DHF (*Bennett et al., 2009*; *Kwon et al., 2008*). Kinetic values are listed in *Figure 1—source data 3*. Based on this calibration, differences between selection coefficients below ~−2.5 were not considered interpretable, and a floor value of −2.5 was applied to all selection coefficients for the purpose of analysis.

For subtraction to calculate Δselection coefficients, null selection coefficients in +Lon selection were substituted with the lowest measured selection coefficient. Mutations with a null selection coefficient in −Lon selection were assigned a Δselection coefficient of 'No data' (colored black). Mutations with 'No data' value in either selection condition were also assigned a Δselection coefficient of 'No data' here.

For clustering of positions, an in-house Python script was used for K-means clustering of positions into categories based on general mutational response at a position (i.e discarding the amino acid identities of the mutants). Spatial clustering was performed based on selection coefficients with the distance between two positions calculated in the following steps: 1) sorting the vectors of selection coefficients for each position, 2) trimming the vectors to match vector lengths after discarding 'no data' values, 3) calculating a Δ vector by subtracting the two sorted and trimmed vectors, and finally calculating the distance as the mean of the absolute value of the Δ vector. For the first round, categories were seeded with virtual positions that have prototypical mutational profiles for the five categories (Beneficial, Tolerant, Mixed, Restricted, and Intolerant). From this first round, all positions in DHFR were categorized into initial clusters. In subsequent rounds, the virtual positions were

removed and candidate positions were compared to the non-self positions populating each cluster. The distance between a candidate position and a cluster of positions is calculated as the average of the distance between the candidate position and the three closest non-self positions in the cluster. Clustering was performed over 10 rounds following the initial seeded round, and convergence was confirmed by observing that five repetitions gave identical clusters.

## Purification of his$_6$-tagged DHFR

DHFR variants were expressed from pHis8 plasmids (KR101/SMT301) for nickel affinity purification as described for one DHFR variant. The plasmid bearing the his-tagged DHFR mutant was transformed via heat shock into chemically competent ER2566 Δ*folA* Δ*thyA* –*Lon* cells, then the cells were plated on LB-agar plates containing 50 µg/mL kanamycin (AMRESCO, cat# 0408, CAS: 25389-94-0, 50 mg/mL in ethanol) and 50 mg/mL thymidine. The plates were incubated overnight at 37 °C. The next day 2 mL of LB medium with 50 µg/mL kanamycin was inoculated with a single colony. This culture was incubated overnight at 37 °C at 225 rpm shaking. The next day, 25 mL of TB medium (12 g Bacto-tryptone, 24 g Bacto-yeast extract, 0.4% glycerol v/v (Sigma Aldrich, cat# G7893, CAS: 56-81-5), brought to 900 mL with MilliQ water, autoclaved, cooled, mixed with 100 mL sterile filtered buffered phosphate (0.17 M KH$_2$PO$_4$ (Sigma Aldrich, cat# P0662, CAS: 7778-77-0), 0.72 M K$_2$HPO$_4$ (Sigma Aldrich, cat# P550, CAS: 16786-57-1))) with 50 µg/mL kanamycin in a 50 mL conical tube was inoculated with 100 µL of the overnight culture. The culture was grown at 37 °C until the OD600 reached 0.5–0.6. Then, the culture was induced with 0.25 mM IPTG (Gold Biotechnology, cat# I2481C100, CAS: 367-93-1, 1M in autoclaved water, sterile filtered) and incubated for 18 hr at 18 °C at 225 rpm shaking. The cultures were pelleted by centrifugation at 3000 rpm for 5 min at 4 °C in a swinging-bucket centrifuge, the supernatant was discarded, and the pellet was resuspended by pipetting in 4 mL/g-pellet of B-PER (ThermoScientific, cat# 78266) with 1 mM PMSF (Millipore Sigma, cat# 7110, CAS: 329-98-6, 100 mM in ethanol), 10 µg/mL leupeptin (VWR Chemicals, cat# J583, CAS: 26305-03-3, 5 mg/mL in water), and 2 µg/mL pepstatin (VWR Chemicals, cat# J580, CAS: 103476-89-7, 2 mg/mL in water). The lysates were incubated at room temperature for 30 min on a rocker and clarified by centrifugation at 3000 rpm for 5 min at 4 °C in a swinging-bucket centrifuge. The lysate supernatant was then transferred to a fresh 50 mL conical tube and incubated for 30 min with 20 µL of NiNTA resin pre-equilibrated in Nickel Binding Buffer (50 mM Tris base (Fisher BioReagents, cat# BP152, CAS: 77-86-1) pH 8.0, 500 mM NaCl, 10 mM imidazole (Fisher Chemical, cat# 03196, CAS: 288-32-4), and then supernatant was removed by pipetting. The resin was washed 3 times for 5 min with 1 mL of Nickel Binding Buffer. Then the protein was eluted into 200 µL of Nickel Elution Buffer (100 mM Tris pH 8.0, 1 M NaCl, 400 mM imidazole) and dialyzed against DHFR Storage Buffer (50 mM Tris pH 8.0, 300 mM NaCl, 1% glycerol v/v) in 3000 Da MW cut-off Slidalyzer dialysis cups (Thermo Scientific, cat# 88401) at 4 °C. After 4 changes of dialysis buffer over 24 hr, the protein was aliquoted, flash frozen in liquid nitrogen, and stored at −80 °C. Proteins were purified to ~90–95% purity as judged from PAGE gel analysis.

## In vitro assay for DHFR velocity and Michaelis-Menten kinetics

In vitro measurements of DHFR velocity were carried out by monitoring the change in UV absorbance. For each mutant screened, a purified enzyme aliquot was thawed and centrifuged at 15,000 rpm for 5 min at 4 °C in a benchtop microcentrifuge. The soluble enzyme was then transferred to a fresh tube, and the concentration was measured by UV absorption on a Nanodrop. Molar concentration of DHFR was calculated using an extinction coefficient of 33585 M$^{-1}$ cm$^{-1}$ at 280 nm for all variants with the following exceptions: 28085 (W30F/M, W47L/M), 35075 (M42Y, R98Y, L165Y), or 39085 (Q102W) M$^{-1}$ cm$^{-1}$. The enzyme was diluted to 555 nM in DHFR storage buffer. A pre-reaction mixture was prepared in MTEN buffer (5 mM MES (Sigma Aldrich, cat# 69889, CAS: 145224-94-8), 25 mM ethanolamine (Sigma Aldrich, cat# E6133, CAS: 2002-24-6), 100 mM NaCl, 25 mM Tris base, pH to 7.0) with 55.5 nM enzyme, 111 µM NADPH (Sigma Aldrich, cat# N7505, CAS: 2646-71-1) and 5 mM DTT (GoldBio, cat# DTT25, CAS: 27565-41-9, 1M in water, sterile filtered). The pre-reaction mixture and a micro quartz cuvette (Fisher Scientific, cat# 14-958-103, 10 mm path length, 2 mm window width) were pre-incubated at 30 °C. The reaction was started by adding 20 µl of 500 µM DHF (Sigma Aldrich, cat# D7006, CAS: 4033-27-6) in MTEN with 5 mM DTT to 180 µL of pre-reaction mixture. The substrate solution was made fresh from a sealed ampule on the day of the

experiment. The reaction was briefly mixed by pipetting and then the reaction was monitored by reading the absorbance at 340 nm with an interval of 0.1 s in a Cary 50 spectrophotometer with the Peltier temperature set to 30 °C. The reactions were allowed to run to completion to establish the baseline, which was subtracted from the absorbance values. The real-time concentration of DHF was calculated by dividing the normalized absorbance values by the decrease in absorbance at 340 nm for the reaction, $0.0132\ \mu M^{-1}cm^{-1}$, the velocity of the reaction was calculated as the slope of linear regression to a 30 s window with a mean DHF concentration equal to 5, 10, 20, or 30 µM. Final velocities were normalized to enzyme concentration.

Michaelis-Menten kinetics were performed as described above using 1–5 µM DHFR for concentrations of DHFR from 0.5 to 100 µM. Initial velocities were estimated from linear regression to the absorbance divided by the decrease in absorbance at 340 nm for the reaction, and then they were fit to the Michaelis-Menten equation using the non-linear least squares method in R.

## Determining DHFR activity and abundance in cell lysates

The cellular activity of DHFR was measured in cell lysates, and then used to calculate DHFR cellular abundance using a method adapted from *Guerrero et al., 2019*; *Rodrigues et al., 2016*. For each characterized DHFR variant, a plasmid (WT DHFR in plasmids SMT102, SMT201, SMT202 and SMT205 with modified promoters and RBSs or DHFR single point mutants in the final selection plasmid SMT205, see The Appendix) was transformed via heat shock into chemically competent *ER2566 ΔfolA ΔthyA ±Lon* cells, which were plated on an LB-agar plate with 30 µg/mL chloramphenicol plus 50 µg/mL thymidine and incubated overnight at 37 °C. On the second day, 2 mL M9 medium with supplements for deficient folate metabolism (50 µg/mL thymidine, 22 µg/mL adenosine, 1 µg/mL calcium pantothenate, 38 µg/mL glycine, and 37.25 µg/mL methionine) and 30 µg/mL chloramphenicol in a 14 ml culture tube was inoculated with a single colony scraped from the plate and incubated at 37 °C at 225 rpm shaking for 12–14 hr. Three biological replicates were obtained from separate single colonies at this step, and all biological replicates were processed in parallel for subsequent steps. All assays were run from fresh transformations. 20–50 µL of the previous culture were used to inoculate 20 mL of M9 medium (no supplements) with 30 µg/mL chloramphenicol in a 50 ml conical tube. This fresh culture was incubated for 12–18 hr at 30 °C at 225 rpm shaking until the OD600 value was between 0.3 and 0.5 on a Cary 50 spectrophotometer over a path of 1 cm. The cultures were pelleted by centrifugation at 3000 rpm for 5 min at 4 °C in a swinging-bucket centrifuge, the supernatant was discarded, and the pellet was thoroughly resuspended in 1.1 mL of M9 medium. 1 mL of the resuspension was transferred to a 1.5 mL Eppendorf tube, and the sample was pelleted at 5000 rpm for 5 min at 4 °C in a microcentrifuge. The supernatant was carefully removed from the pellet, and the pellet was stored at −80 °C until the next step. The remained 100 µL of resuspended pellet was mixed with 900 µL and the OD600 value was measured for each pellet to determine the number of cells in the pellet, with a conversion factor of $8 \times 10^8$ cells/mL at OD600 = 1.0. Pellets for positive (*ER2566*) and negative (*ER2566 ΔfolA ΔthyA ±Lon*) control samples were collected in a similar fashion, except that antibiotics were not used and initial plates were streaked from glycerol stocks. Additionally, the M9 medium for *ER2566 ΔfolA ΔthyA ±Lon* contained folate supplements in every step.

Cell pellets were lysed in B-PER with 1 mM PMSF, 10 µg/mL leupeptin, and 2 µg/mL pepstatin. Volumes for lysis were calculated to have consistent lysate concentration according to the formula: lysis volume = (volume of culture for resuspended pellet)·(OD600 of culture)·(30 µL BPER lysis buffer/1 mL culture). Pellets were resuspended by pipetting in the calculated volume, and the lysates were incubated at room temperature for 30 min on a rocker. The lysates were then clarified by centrifuged at 15,000 rpm for 5 min at 4 °C in a benchtop microcentrifuge. Lysates were kept on ice while the reactions were prepared.

Measurements of DHFR activity in lysates were carried out by monitoring the change in UV absorbance in a BioTek Synergy H1 multimode plate reader. A 180 µL pre-reaction mixture was prepared with MTEN buffer (5 mM MES, 25 mM ethanolamine, 100 mM NaCl, 25 mM Tris base, pH to 7.0), 111 µM NADPH, 5 mM DTT, and containing 20 µL lysate. The pre-reaction mixtures in a UV transparent 96-well plate (Grenier Bio-One, cat# 655809) were pre-incubated at 30 °C for 10 min. The substrate solution of 500 µM DHF in MTEN with 5 mM DTT was made freshly from a sealed ampule of DHF on the day of the experiment. The reaction was started by automatic injection of 20 µl of 500 µM DHF in MTEN with 5 mM DTT into each well with pre-reaction mixture. The plate was then

orbital shaken for 1 min at 365 rpm with a 2 mm amplitude. The reaction was briefly mixed by pipetting and then the reaction was monitored by reading the absorbance at 340 nm with an interval 1 min for 2 hr while incubating at 30 °C. To establish a baseline for accurate calculation of DHF concentration in each well, 50 µL of 1 µL WT DHFR in DHFR storage buffer was injected into each well, the plate was then orbital shaken for 1 min at 365 rpm with a 2 mm amplitude, and the reactions were allowed to run to completion over 10 min, before a final reading of absorbance at 340 nm was taken. In processing, this baseline value was subtracted from the absorbance values for each well. The real-time concentration of DHF was calculated by dividing the normalized absorbance values by the decrease in absorbance at 340 nm for the reaction, $0.0132\ \mu M^{-1}cm^{-1}$, times a correction factor of 1.5 for calibration between the plate reader and the absorbance at 340 nm using a Cary 50 spectrophotometer with a 1 cm pathlength quartz cuvette. The velocity of the reaction was calculated as the slope of linear regression for DHF concentration as a function of time over a window of DHF concentration from 20 to 30 µM. The mean slope of the negative control wells (untransformed *ER2566 ΔfolA ΔthyA ±Lon)* was subtracted from all wells as a baseline. The linear regression of in vitro DHFR reactions using purified enzyme over the same window of DHF concentration from 20 to 30 µM was calculated from measurements described above (section 'In vitro assay for DHFR velocity', *Supplementary file 6*), and the DHFR abundance in each well was calculated from the ratio of activity$_{lysate}$/velocity$_{purified\ enzyme}$. The number of DHFR molecules per cell was then calculated by dividing the total number of DHFR molecules in each 200 µL of reaction by the number of cells in 20 µL of lysate based on the OD600 measurements.

## CD spectroscopy

Samples for circular dichroism (CD) spectroscopy were prepared at a concentration of 10 µM in a buffer of 150 mM NaCl and 50 mM Tris, pH 8.0. CD spectra acquisition and thermal denaturation was carried out in a Jasco J-715 CD spectrometer using a cuvette with a 2 mm pathlength (Starna Cell Inc, cat# 21-Q-2). For each DHFR variant, a pre-denaturation spectra was recorded between 207 nm and 280 nm where the high tension voltage was below 600 V. Thermal denaturation data were collected at 225 nm with a bandwidth of 2 nm, a response time of 8 s, and a resolution of 0.1 ° C during heating at a rate of 1 °C/min. When the curve flattened, the sample was removed from the CD spectrometer and the system was returned to 30 °C. The sample was returned to the chamber and allowed to equilibrate for 10 min. A post-denaturation spectrum was recorded after equilibration. Between samples, the cuvette was cleaned with sonication in Hellmanex III (Hellma, cat# 2805939) followed by washing with 50% concentrated nitric acid. Thermal denaturation was found to be only partially reversible based on comparisons of spectra recorded before and after denaturation. Thermal denaturation curves were fit to a sigmoidal model for the calculation of an approximate apparent $T_m$ for all mutants as previously reported (*Smith et al., 2013*).

## Structural representation of DHFR

All images of the DHFR structure were prepared with UCSF Chimera, and volumetric representations were prepared using the MSMS package (*Sanner et al., 1996*). Solvent accessible surface accessible surface area (SASA) was calculated using the Getarea server (*Fraczkiewicz and Braun, 1998*) for four crystal structures of DHFR (1RX1, 3QL3, 1RX4, and 1RX5) representing different states in DHFR's catalytic cycle. All models were downloaded from PDB_REDO (*Joosten et al., 2014*). For all positions in DHFR, if the residue had <20% SASA in any structure, the residue was classified as buried. All other residues were classified as exposed. Burial classification is reported in *Figure 3— source data 1*.

The distance between the positions within each mutational response category and sites within the DHFR structure (hydride transfer site, M20 loop, core of the globular domain, and the beta-sheet surface beneath the active site) were determined using a model of the transition state provided by Phil Hanoian (*Liu et al., 2013*). The representative atom for the hydride transfer site is the hydride atom in the transition state model. The representative atom for the adenine ring is C5 (C18 in the pdb). The representative atom for the core of the globular domain is the alpha carbon of I41. The representative atom for the beta sheet region is the alpha carbon of D114. For all cases, the distance is defined as the distance between the representative atom and the alpha carbon of the target position.

Mean atom neighbors for each residue on a structure were calculated using an in-house python script. The number of non-hydrogen atoms within an 8 Å shell of each non-hydrogen atom in the structure were counted and averaged for all non-hydrogen atoms at each side chain. These values we calculated for four crystal structures of DHFR (PDB IDs: 1RX1, 3QL3, 1RX4, 1RX5) and averaged over the set.

### Profile similarity analysis

We downloaded the DHFR alignment from OpenSeq.org (*Ovchinnikov et al., 2014*), selected all bacterial DHFR sequences, and aligned the *E. coli* DHFR sequence to the MSA using MUSCLE (*Edgar, 2004*). This multiple sequence alignment (MSA) is provided in *Supplementary file 7*. Frequencies for each amino acid at each sequence position in the MSA were calculated from counts in each column, with absent amino acids given an arbitrarily low frequency of 0.0001. To compare the amino acid frequencies from the MSA to the selection coefficients, we first divided the selection coefficients by ln(2) • 18 hr to convert from the Enrich selection coefficients to a Δdoubling rate. We then multiplied the Δdoubling rate by −1 and back-calculated frequencies using Boltzmann weighting using a temperature (0.44 kT for –Lon selection, and 0.47 kT for +Lon selection) that resulted in the mean sequence entropy to be within ±0.01 of that of the MSA (0.50). Then, profile similarity at each sequence position was calculated as 1 – the Jensen-Shannon Divergence of the amino acid frequencies. Profile similarity was determined over columns corresponding to positions 2–158 because the DHFR library begins at position two and the DHFR MSA cuts off after position 158.

## Acknowledgements

The authors would like to thank Carol Gross, Melanie Silvis, and Byoung Mo Koo for discussion and for providing Lambda red plasmids; Rama Ranganathan and Victor Salinas for supplying parts and expertise for the construction of the turbidostat; Sharon Hammes-Shiffer and Phil Hanoian for providing QM/MM models of the hydride transfer step; Natasha Carli and Jim McGuire at the Gladstone Institute Genomics Core for performing NextSeq 500 sequencing runs with support from the James B Pendleton Charitable Trust; and Norma Neff, Anna Sellas, and Rene Sit for performing and aiding Miseq and NextSeq 500 sequencing runs at the Chan Zuckerberg Biohub.

## Additional information

### Funding

| Funder | Grant reference number | Author |
| --- | --- | --- |
| National Science Foundation | MCB 1615990 | Tanja Kortemme |
| Gordon and Betty Moore Foundation | GBMF4557 | Kimberly A Reynolds |
| National Science Foundation | Graduate Student Fellowship | Samuel Thompson |
| UCSF Chuan Lyu Chancellor's Fellowship | Graduate Student Fellowship | Samuel Thompson |

The funders had no role in study design, data collection and interpretation, or the decision to submit the work for publication.

### Author contributions

Samuel Thompson, Conceptualization, Software, Formal analysis, Funding acquisition, Investigation, Visualization, Methodology, Writing - original draft, Writing - review and editing; Yang Zhang, Formal analysis, Investigation, Writing - review and editing; Christine Ingle, Investigation, Methodology, Writing - review and editing; Kimberly A Reynolds, Resources, Formal analysis, Supervision, Funding acquisition, Project administration, Writing - review and editing; Tanja Kortemme, Conceptualization, Resources, Formal analysis, Supervision, Funding acquisition, Writing - original draft, Project administration, Writing - review and editing

**Author ORCIDs**
Samuel Thompson  https://orcid.org/0000-0001-6468-9538
Christine Ingle  http://orcid.org/0000-0002-0203-2845
Kimberly A Reynolds  https://orcid.org/0000-0003-4805-0317
Tanja Kortemme  https://orcid.org/0000-0002-8494-680X

**Decision letter and Author response**
Decision letter https://doi.org/10.7554/eLife.53476.sa1
Author response https://doi.org/10.7554/eLife.53476.sa2

## Additional files

**Supplementary files**

• Supplementary file 1. Selection coefficients for –Lon selection measured as described in Materials and methods are reported with the standard deviation between biological replicates and the standard error from linear regression (as calculated by Enrich2; *Rubin et al., 2017*). Values are reported as calculated, but based on the selection calibration, differences between selection coefficients with values below ~–2.5 are not interpretable.

• Supplementary file 2. Selection coefficients for +Lon selection measured as described in Materials and methods are reported with the standard deviation between biological replicates and the standard error from linear regression. Values are reported as calculated, but based on the selection calibration, differences between selection coefficients with values below ~–2.5 are not interpretable.

• Supplementary file 3. Raw deep sequencing counts for the calibration set of mutants –Lon selection. Counts are recorded for all turbidostat timepoints over three repeats.

• Supplementary file 4. Raw deep sequencing counts for single point mutants in –Lon selection. Counts are recorded for a sample collected from the transformation rescue medium, a sample from overnight outgrowth in supplemented M9, and for all turbidostat timepoints over six experiments. In each experiment, 2 of 4 sublibraries were screened as described in Materials and methods, for a total of 3 repeats over the full library.

• Supplementary file 5. Raw deep sequencing counts for single point mutants in +Lon selection. Counts are recorded as in *Supplementary file 4*.

• Supplementary file 6. DHFR reaction velocities as a function of DHF concentration used for the measurement of soluble DHFR abundance from lysate activities as described in Materials and methods. For each mutant (column 1), three technical repeats are included (column 2). There are two lines for each repeat reaction, one for DHF concentration and one for the reaction velocity at that concentration (column 3). All data in columns 4+ are the experimental values.

• Supplementary file 7. Multiple sequence alignment of bacterial DHFR sequences generated as described in Materials and methods and used for bioinformatics analyses.

• Transparent reporting form

### Data availability

Source data have been provided for Figure 1 (Figure 1—source data 1–3, Supplementary file 1), Figure 2 (Supplementary files 1 and 2), Figure 3 (Figure 3—source data 1, Supplementary files 1 and 2), Figure 4 (Figure 4—source data 1–3, Supplementary file 7) and Figure 5 (Figure 3—source data 1). Code for analysis is available in our GitHub repository for this project (https://github.com/keleayon/2019_DHFR_Lon.git; copy archived at https://github.com/elifesciences-publications/2019_DHFR_Lon) along with key input files and example command lines. Raw deep sequencing data was deposited to the Sequence Read Archive in entry PRJNA590072 (BioSamples: SAMN13316587, SAMN13316662). Allele counts used to generate the selection coefficients (all figures) are reported in Supplementary files 4–6. Key plasmids (Appendix 1) will be available from Addgene.

The following dataset was generated:

| Author(s) | Year | Dataset title | Dataset URL | Database and Identifier |
|---|---|---|---|---|
| Thompson S | 2019 | Mapping the mutational landscape of DHFR single point mutants with perturbations to the cellular environment | https://www.ncbi.nlm.nih.gov/bioproject/?term=PRJNA590072 | NCBI BioProject, PRJNA590072 |

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

## Appendix 1

**Appendix 1—key resources table**

| Reagent type (species) or resource | Designation | Source or reference | Identifiers | Additional information |
|---|---|---|---|---|
| Strain, strain background (*Escherichia coli*) | ER2566 | New England Biolabs | Cat# C2566I | Chemically competent cells |
| Strain, strain background (*Escherichia coli*) | ER2566 Δ*folA*/Δ*thyA* (–Lon) | Reynolds et al. Cell 2011 | | Chemically competent and electrocompetent cells |
| Strain, strain background (*Escherichia coli*) | ER2566 Δ*folA*/Δ*thyA* (+Lon) | This work | | Chemically competent and electrocompetent cells |
| Recombinant DNA reagent | SMT101 (plasmid) | This work | | Dual expression of DHFR and TYMS, in vivo assays, chloramphenicol (35 µg/mL final concentration) |
| Recombinant DNA reagent | SMT201 (plasmid) | This work | | SMT101 with TET promter for TYMS, in vivo assays, Chloramphenicol (35 µg/mL final concentration) |
| Recombinant DNA reagent | SMT205 (plasmid) | This work | | SMT201 with mutated RBS for DHFR, in vivo assays, Chloramphenicol (35 µg/mL final concentration) |
| Recombinant DNA reagent | SMT215 (plasmid) | This work | | SMT205 with DHFR-FLAG-tag, western blot, Chloramphenicol (35 µg/mL final concentration) |
| Recombinant DNA reagent | KR101/SMT301 (plasmid) | Reynolds et al. Cell 2011 | | His8-tag, Heterologous expression, NiNTA purfication, kanamycin (50 µg/mL final concentration) |
| Recombinant DNA reagent | pSIM6 (plasmid) | *Blomfield et al., 1991* | | Lambda Red recombinase expression, temperature-sensitive promoter, ampicillin/carbenicilin (100 µg/mL final concentration) |
| Recombinant DNA reagent | pIB279 (plasmid) | *Blomfield et al., 1991* | | KAN-SacB cassette for positive/ negative selection, ampicillin/ carbenicilin (100 µg/mL final concentration) |
| Sequence-based reagent | TetDuet1_sense | This work | Mutagenic PCR primer | ccgCTTAAGtcgaacagaaagt aatcgtattgtacatccctatc |
| Sequence-based reagent | TetDuet2_anti | This work | Mutagenic PCR primer | gatagggatgtcaatctctatcact gatagggatgtacaatacg |
| Sequence-based reagent | TetDuet3_sense | This work | Mutagenic PCR primer | agagattgacatccctatcagtgat agagatactgagcacatcag |
| Sequence-based reagent | TetDuet4_anti | This work | Mutagenic PCR primer | ctttaatgaattcggtcagtgcgtcct gctgatgtgctcagtatctc |

*Appendix 1—key resources table continued*

| Reagent type (species) or resource | Designation | Source or reference | Identifiers | Additional information |
|---|---|---|---|---|
| Sequence-based reagent | TetDuet5_sense | This work | Mutagenic PCR primer | cactgaccgaattcattaaagaggag aaaggtaccatatggc |
| Sequence-based reagent | TetDuet_5flanking | This work | Mutagenic PCR primer | ccgcttaagtcgaacagaaag |
| Sequence-based reagent | TetDuet_3flanking | This work | Mutagenic PCR primer | cggagatctgccatatggtacc |
| Sequence-based reagent | WT_DHFR_pos2_fwd | This work | Mutagenic PCR primer | NNSAGTCTGATTGCGGCGTTAG |
| Sequence-based reagent | WT_DHFR_pos2_fwd2 | This work | Mutagenic PCR primer | NNSAGTCTGATTGCGGCGTTAG |
| Sequence-based reagent | WT_DHFR_pos3_fwd | This work | Mutagenic PCR primer | NNSCTGATTGCGGCGTTAGCG |
| Sequence-based reagent | WT_DHFR_pos4_fwd | This work | Mutagenic PCR primer | NNSATTGCGGCGTTAGCGGTA |
| Sequence-based reagent | WT_DHFR_pos5_fwd | This work | Mutagenic PCR primer | NNSGCGGCGTTAGCGGTAGAT |
| Sequence-based reagent | WT_DHFR_pos6_fwd | This work | Mutagenic PCR primer | NNSGCGTTAGCGGTAGATCGC |
| Sequence-based reagent | WT_DHFR_pos7_fwd | This work | Mutagenic PCR primer | NNSTTAGCGGTAGATCGCGTTA TC |
| Sequence-based reagent | WT_DHFR_pos8_fwd | This work | Mutagenic PCR primer | NNSGCGGTAGATCGCGTTA TCG |
| Sequence-based reagent | WT_DHFR_pos8_fwd2 | This work | Mutagenic PCR primer | NNSGCGGTAGATCGCGTTA TCG |
| Sequence-based reagent | WT_DHFR_pos9_fwd | This work | Mutagenic PCR primer | NNSGTAGATCGCGTTATCGGCA TG |
| Sequence-based reagent | WT_DHFR_pos10_fwd | This work | Mutagenic PCR primer | NNSGATCGCGTTATCGGCA TGG |
| Sequence-based reagent | WT_DHFR_pos11_fwd | This work | Mutagenic PCR primer | NNSCGCGTTATCGGCA TGGAAAA |
| Sequence-based reagent | WT_DHFR_pos12_fwd | This work | Mutagenic PCR primer | NNSGTTATCGGCA TGGAAAACGC |
| Sequence-based reagent | WT_DHFR_pos13_fwd | This work | Mutagenic PCR primer | NNSATCGGCATGGAAAACGCC |
| Sequence-based reagent | WT_DHFR_pos14_fwd | This work | Mutagenic PCR primer | NNSGGCATGGAAAACGCCATG |

*Appendix 1—key resources table continued*

| Reagent type (species) or resource | Designation | Source or reference | Identifiers | Additional information |
|---|---|---|---|---|
| Sequence-based reagent | WT_DHFR_pos15_fwd | This work | Mutagenic PCR primer | NNSATGGAAAACGCCATGCCG |
| Sequence-based reagent | WT_DHFR_pos16_fwd | This work | Mutagenic PCR primer | NNSGAAAACGCCATGCCGTGG |
| Sequence-based reagent | WT_DHFR_pos17_fwd | This work | Mutagenic PCR primer | NNSAACGCCATGCCGTGGAAC |
| Sequence-based reagent | WT_DHFR_pos18_fwd | This work | Mutagenic PCR primer | NNSGCCATGCCGTGGAACCTG |
| Sequence-based reagent | WT_DHFR_pos19_fwd | This work | Mutagenic PCR primer | NNSATGCCGTGGAACCTGCCT |
| Sequence-based reagent | WT_DHFR_pos20_fwd | This work | Mutagenic PCR primer | NNSCCGTGGAACCTGCCTGCC |
| Sequence-based reagent | WT_DHFR_pos21_fwd | This work | Mutagenic PCR primer | NNSTGGAACCTGCCTGCCGAT |
| Sequence-based reagent | WT_DHFR_pos22_fwd | This work | Mutagenic PCR primer | NNSAACCTGCCTGCCGATCTC |
| Sequence-based reagent | WT_DHFR_pos22_fwd2 | This work | Mutagenic PCR primer | NNSAACCTGCCTGCCGATCTC |
| Sequence-based reagent | WT_DHFR_pos23_fwd | This work | Mutagenic PCR primer | NNSCTGCCTGCCGATCTCGCC |
| Sequence-based reagent | WT_DHFR_pos24_fwd | This work | Mutagenic PCR primer | NNSCCTGCCGATCTCGCCTGG |
| Sequence-based reagent | WT_DHFR_pos25_fwd | This work | Mutagenic PCR primer | NNSGCCGATCTCGCCTGGTTT |
| Sequence-based reagent | WT_DHFR_pos26_fwd | This work | Mutagenic PCR primer | NNSGATCTCGCCTGG TTTAAACGC |
| Sequence-based reagent | WT_DHFR_pos27_fwd | This work | Mutagenic PCR primer | NNSCTCGCCTGGTTTAAACG-CAACA |
| Sequence-based reagent | WT_DHFR_pos28_fwd | This work | Mutagenic PCR primer | NNSGCCTGGTTTAAACGCAA-CAC |
| Sequence-based reagent | WT_DHFR_pos29_fwd | This work | Mutagenic PCR primer | NNSTGGTTTAAACGCAACACCT TAAATAAAC |
| Sequence-based reagent | WT_DHFR_pos30_fwd | This work | Mutagenic PCR primer | NNSTTTAAACGCAACACC TTAAAT AAACCCG |
| Sequence-based reagent | WT_DHFR_pos31_fwd | This work | Mutagenic PCR primer | NNSAAACGCAACACCTTAAA TAA ACCCGTG |

*Appendix 1—key resources table continued*

| Reagent type (species) or resource | Designation | Source or reference | Identifiers | Additional information |
|---|---|---|---|---|
| Sequence-based re-agent | WT_DHFR_pos32_fwd | This work | Mutagenic PCR primer | NNSCGCAACACCTTAAA TAAACCCGT |
| Sequence-based re-agent | WT_DHFR_pos33_fwd | This work | Mutagenic PCR primer | NNSAACACCTTAAATAAACCCG TGA TTATGG |
| Sequence-based re-agent | WT_DHFR_pos34_fwd | This work | Mutagenic PCR primer | NNSACCTTAAATAAACCCGTGA TTATGGG |
| Sequence-based re-agent | WT_DHFR_pos35_fwd | This work | Mutagenic PCR primer | NNSTTAAATAAACCCGTGATTA TGGGCC |
| Sequence-based re-agent | WT_DHFR_pos36_fwd | This work | Mutagenic PCR primer | NNSAATAAACCCGTGATTAT GGGCC |
| Sequence-based re-agent | WT_DHFR_pos37_fwd | This work | Mutagenic PCR primer | NNSAAACCCGTGATTATGGGCC |
| Sequence-based re-agent | WT_DHFR_pos38_fwd | This work | Mutagenic PCR primer | NNSCCCGTGATTATGGGCCGC |
| Sequence-based re-agent | WT_DHFR_pos39_fwd | This work | Mutagenic PCR primer | NNSGTGATTATGGGCCGCCA TAC |
| Sequence-based re-agent | WT_DHFR_pos40_fwd | This work | Mutagenic PCR primer | NNSATTATGGGCCGCCATACCT |
| Sequence-based re-agent | WT_DHFR_pos41_fwd | This work | Mutagenic PCR primer | NNSATGGGCCGCCATACCTGG |
| Sequence-based re-agent | WT_DHFR_pos42_fwd | This work | Mutagenic PCR primer | NNSGGCCGCCATACCTGGGAA |
| Sequence-based re-agent | WT_DHFR_pos42_fwd2 | This work | Mutagenic PCR primer | NNSGGCCGCCATACCTGGGAA TC |
| Sequence-based re-agent | WT_DHFR_pos43_fwd | This work | Mutagenic PCR primer | NNSCGCCATACCTGGGAATCAA TC |
| Sequence-based re-agent | WT_DHFR_pos43_fwd2 | This work | Mutagenic PCR primer | NNSCGCCATACCTGGGAATCAA TC |
| Sequence-based re-agent | WT_DHFR_pos44_fwd | This work | Mutagenic PCR primer | NNSCATACCTGGGAATCAA TCGGTC |
| Sequence-based re-agent | WT_DHFR_pos45_fwd | This work | Mutagenic PCR primer | NNSACCTGGGAATCAATCGGTC |
| Sequence-based re-agent | WT_DHFR_pos46_fwd | This work | Mutagenic PCR primer | NNSTGGGAATCAATCGGTCGTC |
| Sequence-based re-agent | WT_DHFR_pos47_fwd | This work | Mutagenic PCR primer | NNSGAATCAATCGGTCGTCCG TTG |

*Appendix 1—key resources table continued*

| Reagent type (species) or resource | Designation | Source or reference | Identifiers | Additional information |
|---|---|---|---|---|
| Sequence-based reagent | WT_DHFR_pos48_fwd | This work | Mutagenic PCR primer | NNSTCAATCGGTCGTCCGTTGC |
| Sequence-based reagent | WT_DHFR_pos49_fwd | This work | Mutagenic PCR primer | NNSATCGGTCGTCCGTTGCCA |
| Sequence-based reagent | WT_DHFR_pos50_fwd | This work | Mutagenic PCR primer | NNSGGTCGTCCGTTGCCAG-GAC |
| Sequence-based reagent | WT_DHFR_pos51_fwd | This work | Mutagenic PCR primer | NNSCGTCCGTTGCCAGGACGC |
| Sequence-based reagent | WT_DHFR_pos52_fwd | This work | Mutagenic PCR primer | NNSCCGTTGCCAGGACGCAAA |
| Sequence-based reagent | WT_DHFR_pos53_fwd | This work | Mutagenic PCR primer | NNSTTGCCAGGACGCAAAAA TATTATCC |
| Sequence-based reagent | WT_DHFR_pos54_fwd | This work | Mutagenic PCR primer | NNSCCAGGACGCAAAAATATT ATCCTCAG |
| Sequence-based reagent | WT_DHFR_pos55_fwd | This work | Mutagenic PCR primer | NNSGGACGCAAAAATATTATC CTCAGCAG |
| Sequence-based reagent | WT_DHFR_pos56_fwd | This work | Mutagenic PCR primer | NNSCGCAAAAATATTATCCTCA GCAGTCAA |
| Sequence-based reagent | WT_DHFR_pos57_fwd | This work | Mutagenic PCR primer | NNSAAAAATATTATCCTCAGCA GTCAACCGG |
| Sequence-based reagent | WT_DHFR_pos58_fwd | This work | Mutagenic PCR primer | NNSAATATTATCCTCAGCAGTC AACCGGGTA |
| Sequence-based reagent | WT_DHFR_pos59_fwd | This work | Mutagenic PCR primer | NNSATTATCCTCAGCAG TCAACCG |
| Sequence-based reagent | WT_DHFR_pos60_fwd | This work | Mutagenic PCR primer | NNSATCCTCAGCAGTCAACCG |
| Sequence-based reagent | WT_DHFR_pos61_fwd | This work | Mutagenic PCR primer | NNSCTCAGCAGTCAACCGGGT |
| Sequence-based reagent | WT_DHFR_pos62_fwd | This work | Mutagenic PCR primer | NNSAGCAGTCAACCGGGTACG |
| Sequence-based reagent | WT_DHFR_pos63_fwd | This work | Mutagenic PCR primer | NNSAGTCAACCGGGTACGGAC |
| Sequence-based reagent | WT_DHFR_pos64_fwd | This work | Mutagenic PCR primer | NNSCAACCGGGTACGGACGAT |
| Sequence-based reagent | WT_DHFR_pos65_fwd | This work | Mutagenic PCR primer | NNSCCGGGTACGGACGATCGC |

*Appendix 1—key resources table continued*

| Reagent type (species) or resource | Designation | Source or reference | Identifiers | Additional information |
|---|---|---|---|---|
| Sequence-based reagent | WT_DHFR_pos66_fwd | This work | Mutagenic PCR primer | NNSGGTACGGACGATCGCGTA |
| Sequence-based reagent | WT_DHFR_pos66_fwd2 | This work | Mutagenic PCR primer | NNSGGTACGGACGATCGCGTAAC |
| Sequence-based reagent | WT_DHFR_pos67_fwd | This work | Mutagenic PCR primer | NNSACGGACGATCGCGTAACG |
| Sequence-based reagent | WT_DHFR_pos67_fwd2 | This work | Mutagenic PCR primer | NNSACGGACGATCGCGTAACG |
| Sequence-based reagent | WT_DHFR_pos68_fwd | This work | Mutagenic PCR primer | NNSGACGATCGCGTAACGTGG |
| Sequence-based reagent | WT_DHFR_pos69_fwd | This work | Mutagenic PCR primer | NNSGATCGCGTAACGTGGGTG |
| Sequence-based reagent | WT_DHFR_pos70_fwd | This work | Mutagenic PCR primer | NNSCGCGTAACGTGGGTGAAG |
| Sequence-based reagent | WT_DHFR_pos71_fwd | This work | Mutagenic PCR primer | NNSGTAACGTGGGTGAAGTCGG |
| Sequence-based reagent | WT_DHFR_pos72_fwd | This work | Mutagenic PCR primer | NNSACGTGGGTGAAGTCGGTG |
| Sequence-based reagent | WT_DHFR_pos73_fwd | This work | Mutagenic PCR primer | NNSTGGGTGAAGTCGGTGGAT |
| Sequence-based reagent | WT_DHFR_pos73_fwd2 | This work | Mutagenic PCR primer | NNSTGGGTGAAGTCGGTGGATG |
| Sequence-based reagent | WT_DHFR_pos74_fwd | This work | Mutagenic PCR primer | NNSGTGAAGTCGGTGGATGAAGC |
| Sequence-based reagent | WT_DHFR_pos74_fwd2 | This work | Mutagenic PCR primer | NNSGTGAAGTCGGTGGATGAAGC |
| Sequence-based reagent | WT_DHFR_pos75_fwd | This work | Mutagenic PCR primer | NNSAAGTCGGTGGATGAAGCCAT |
| Sequence-based reagent | WT_DHFR_pos76_fwd | This work | Mutagenic PCR primer | NNSTCGGTGGATGAAGCCATC |
| Sequence-based reagent | WT_DHFR_pos77_fwd | This work | Mutagenic PCR primer | NNSGTGGATGAAGCCATCGCG |
| Sequence-based reagent | WT_DHFR_pos78_fwd | This work | Mutagenic PCR primer | NNSGATGAAGCCATCGCGGCG |
| Sequence-based reagent | WT_DHFR_pos79_fwd | This work | Mutagenic PCR primer | NNSGAAGCCATCGCGGCGTGT |

*Appendix 1—key resources table continued*

| Reagent type (species) or resource | Designation | Source or reference | Identifiers | Additional information |
|---|---|---|---|---|
| Sequence-based reagent | WT_DHFR_pos80_fwd | This work | Mutagenic PCR primer | NNSGCCATCGCGGCGTGTGGT |
| Sequence-based reagent | WT_DHFR_pos80_fwd2 | This work | Mutagenic PCR primer | NNSGCCATCGCGGCGTGTGG |
| Sequence-based reagent | WT_DHFR_pos81_fwd | This work | Mutagenic PCR primer | NNSATCGCGGCGTGTGGTGAC |
| Sequence-based reagent | WT_DHFR_pos82_fwd | This work | Mutagenic PCR primer | NNSGCGGCGTGTGGTGACGTA |
| Sequence-based reagent | WT_DHFR_pos82_fwd2 | This work | Mutagenic PCR primer | NNSGCGGCGTGTGGTGACGTACCAGAAATC |
| Sequence-based reagent | WT_DHFR_pos83_fwd | This work | Mutagenic PCR primer | NNSGCGTGTGGTGACGTACCA |
| Sequence-based reagent | WT_DHFR_pos84_fwd | This work | Mutagenic PCR primer | NNSTGTGGTGACGTACCA-GAAATCAT |
| Sequence-based reagent | WT_DHFR_pos84_fwd2 | This work | Mutagenic PCR primer | NNSTGTGGTGACGTACCA-GAAATCATG |
| Sequence-based reagent | WT_DHFR_pos85_fwd | This work | Mutagenic PCR primer | NNSGGTGACGTACCAGAAATCATGG |
| Sequence-based reagent | WT_DHFR_pos86_fwd | This work | Mutagenic PCR primer | NNSGACGTACCAGAAATCATGGTGATTGG |
| Sequence-based reagent | WT_DHFR_pos87_fwd | This work | Mutagenic PCR primer | NNSGTACCAGAAATCATGGTGATTGGCGG |
| Sequence-based reagent | WT_DHFR_pos88_fwd | This work | Mutagenic PCR primer | NNSCCAGAAATCATGGTGATTGGCGG |
| Sequence-based reagent | WT_DHFR_pos89_fwd | This work | Mutagenic PCR primer | NNSGAAATCATGGTGATTGGCGGCG |
| Sequence-based reagent | WT_DHFR_pos89_fwd2 | This work | Mutagenic PCR primer | NNSGAAATCATGGTGATTGGCGGC |
| Sequence-based reagent | WT_DHFR_pos90_fwd | This work | Mutagenic PCR primer | NNSATCATGGTGATTGGCGGC |
| Sequence-based reagent | WT_DHFR_pos91_fwd | This work | Mutagenic PCR primer | NNSATGGTGATTGGCGGCGGTC |
| Sequence-based reagent | WT_DHFR_pos92_fwd | This work | Mutagenic PCR primer | NNSGTGATTGGCGGCGGTCGC |
| Sequence-based reagent | WT_DHFR_pos93_fwd | This work | Mutagenic PCR primer | NNSATTGGCGGCGGTCGCGTTTA |

*Appendix 1—key resources table continued*

| Reagent type (species) or resource | Designation | Source or reference | Identifiers | Additional information |
|---|---|---|---|---|
| Sequence-based re-agent | WT_DHFR_pos94_fwd | This work | Mutagenic PCR primer | NNSGGCGGCGGTCGCGTTTAT |
| Sequence-based re-agent | WT_DHFR_pos95_fwd | This work | Mutagenic PCR primer | NNSGGCGGTCGCGTTTATGAA |
| Sequence-based re-agent | WT_DHFR_pos95_fwd2 | This work | Mutagenic PCR primer | NNSGGCGGTCGCGTTTATGAAC |
| Sequence-based re-agent | WT_DHFR_pos96_fwd | This work | Mutagenic PCR primer | NNSGGTCGCGTTTATGAACAGTTCTT |
| Sequence-based re-agent | WT_DHFR_pos97_fwd | This work | Mutagenic PCR primer | NNSCGCGTTTATGAACAGTTCTTGC |
| Sequence-based re-agent | WT_DHFR_pos98_fwd | This work | Mutagenic PCR primer | NNSGTTTATGAACAGTTCTTGCCAAAAGCGC |
| Sequence-based re-agent | WT_DHFR_pos99_fwd | This work | Mutagenic PCR primer | NNSTATGAACAGTTCTTGCCAAAAGCGC |
| Sequence-based re-agent | WT_DHFR_pos100_fwd | This work | Mutagenic PCR primer | NNSGAACAGTTCTTGCCAAAAGCGCAAAAAC |
| Sequence-based re-agent | WT_DHFR_pos101_fwd | This work | Mutagenic PCR primer | NNSCAGTTCTTGCCAAAAGCG-CAAAAAC |
| Sequence-based re-agent | WT_DHFR_pos102_fwd | This work | Mutagenic PCR primer | NNSTTCTTGCCAAAAGCG-CAAAAAC |
| Sequence-based re-agent | WT_DHFR_pos103_fwd | This work | Mutagenic PCR primer | NNSTTGCCAAAAGCGCAAAAACTGTAT |
| Sequence-based re-agent | WT_DHFR_pos104_fwd | This work | Mutagenic PCR primer | NNSCCAAAAGCGCAAAAACTGTATCTGA |
| Sequence-based re-agent | WT_DHFR_pos104_fwd2 | This work | Mutagenic PCR primer | NNSCCAAAAGCGCAAAAACTGTATCTG |
| Sequence-based re-agent | WT_DHFR_pos105_fwd | This work | Mutagenic PCR primer | NNSAAAGCGCAAAAACTGTATCTGACG |
| Sequence-based re-agent | WT_DHFR_pos106_fwd | This work | Mutagenic PCR primer | NNSGCGCAAAAACTGTATCTGACG |
| Sequence-based re-agent | WT_DHFR_pos107_fwd | This work | Mutagenic PCR primer | NNSCAAAAACTGTATCTGACG-CATATCGAC |
| Sequence-based re-agent | WT_DHFR_pos107_fwd2 | This work | Mutagenic PCR primer | NNSCAAAAACTGTATCTGACG-CATATCG |
| Sequence-based re-agent | WT_DHFR_pos108_fwd | This work | Mutagenic PCR primer | NNSAAACTGTATCTGACGCATATCGAC |

*Appendix 1—key resources table continued*

| Reagent type (species) or resource | Designation | Source or reference | Identifiers | Additional information |
|---|---|---|---|---|
| Sequence-based reagent | WT_DHFR_pos109_fwd | This work | Mutagenic PCR primer | NNSCTGTATCTGACGCATA TCGACG |
| Sequence-based reagent | WT_DHFR_pos110_fwd | This work | Mutagenic PCR primer | NNSTATCTGACGCATA TCGACGCA |
| Sequence-based reagent | WT_DHFR_pos111_fwd | This work | Mutagenic PCR primer | NNSCTGACGCATATCGACG-CAG |
| Sequence-based reagent | WT_DHFR_pos112_fwd | This work | Mutagenic PCR primer | NNSACGCATATCGACGCA-GAAGT |
| Sequence-based reagent | WT_DHFR_pos113_fwd | This work | Mutagenic PCR primer | NNSCATATCGACGCAGAAG TGGAAG |
| Sequence-based reagent | WT_DHFR_pos114_fwd | This work | Mutagenic PCR primer | NNSATCGACGCAGAAG TGGAAG |
| Sequence-based reagent | WT_DHFR_pos115_fwd | This work | Mutagenic PCR primer | NNSGACGCAGAAGTGGAAGGC |
| Sequence-based reagent | WT_DHFR_pos116_fwd | This work | Mutagenic PCR primer | NNSGCAGAAGTGGAAGGCGAC |
| Sequence-based reagent | WT_DHFR_pos117_fwd | This work | Mutagenic PCR primer | NNSGAAGTGGAAGGCGACACC |
| Sequence-based reagent | WT_DHFR_pos118_fwd | This work | Mutagenic PCR primer | NNSGTGGAAGGCGACACCCAT |
| Sequence-based reagent | WT_DHFR_pos118_fwd2 | This work | Mutagenic PCR primer | NNSGTGGAAGGCGACACCCA TTTC |
| Sequence-based reagent | WT_DHFR_pos119_fwd | This work | Mutagenic PCR primer | NNSGAAGGCGACACCCA TTTCC |
| Sequence-based reagent | WT_DHFR_pos120_fwd | This work | Mutagenic PCR primer | NNSGGCGACACCCATTTCCCG |
| Sequence-based reagent | WT_DHFR_pos121_fwd | This work | Mutagenic PCR primer | NNSGACACCCATTTCCCGGA TTAC |
| Sequence-based reagent | WT_DHFR_pos122_fwd | This work | Mutagenic PCR primer | NNSACCCATTTCCCGGATTAC-GA |
| Sequence-based reagent | WT_DHFR_pos123_fwd | This work | Mutagenic PCR primer | NNSCATTTCCCGGATTAC-GAGCC |
| Sequence-based reagent | WT_DHFR_pos124_fwd | This work | Mutagenic PCR primer | NNSTTCCCGGATTACGAGCCG |
| Sequence-based reagent | WT_DHFR_pos125_fwd | This work | Mutagenic PCR primer | NNSCCGGATTACGAGCCGGAT |

*Appendix 1—key resources table continued*

| Reagent type (species) or resource | Designation | Source or reference | Identifiers | Additional information |
|---|---|---|---|---|
| Sequence-based reagent | WT_DHFR_pos126_fwd | This work | Mutagenic PCR primer | NNSGATTACGAGCCGGATGACTG |
| Sequence-based reagent | WT_DHFR_pos127_fwd | This work | Mutagenic PCR primer | NNSTACGAGCCGGATGACTGG |
| Sequence-based reagent | WT_DHFR_pos128_fwd | This work | Mutagenic PCR primer | NNSGAGCCGGATGACTGGGAA |
| Sequence-based reagent | WT_DHFR_pos129_fwd | This work | Mutagenic PCR primer | NNSCCGGATGACTGGGAATCG |
| Sequence-based reagent | WT_DHFR_pos130_fwd | This work | Mutagenic PCR primer | NNSGATGACTGGGAATCGGTATTCAG |
| Sequence-based reagent | WT_DHFR_pos131_fwd | This work | Mutagenic PCR primer | NNSGACTGGGAATCGGTATTCAGC |
| Sequence-based reagent | WT_DHFR_pos131_fwd2 | This work | Mutagenic PCR primer | NNSGACTGGGAATCGGTATTCAGC |
| Sequence-based reagent | WT_DHFR_pos132_fwd | This work | Mutagenic PCR primer | NNSTGGGAATCGGTATTCAGC-GAATT |
| Sequence-based reagent | WT_DHFR_pos133_fwd | This work | Mutagenic PCR primer | NNSGAATCGGTATTCAGCGAATTCCAC |
| Sequence-based reagent | WT_DHFR_pos134_fwd | This work | Mutagenic PCR primer | NNSTCGGTATTCAGCGAATTCCAC |
| Sequence-based reagent | WT_DHFR_pos135_fwd | This work | Mutagenic PCR primer | NNSGTATTCAGCGAATTCCAC-GATG |
| Sequence-based reagent | WT_DHFR_pos135_fwd2 | This work | Mutagenic PCR primer | NNSGTATTCAGCGAATTCCAC-GATGC |
| Sequence-based reagent | WT_DHFR_pos136_fwd | This work | Mutagenic PCR primer | NNSTTCAGCGAATTCCACGATGC |
| Sequence-based reagent | WT_DHFR_pos136_fwd2 | This work | Mutagenic PCR primer | NNSTTCAGCGAATTCCACGATGC |
| Sequence-based reagent | WT_DHFR_pos137_fwd | This work | Mutagenic PCR primer | NNSAGCGAATTCCACGATGCTG |
| Sequence-based reagent | WT_DHFR_pos138_fwd | This work | Mutagenic PCR primer | NNSGAATTCCACGATGCTGATGC |
| Sequence-based reagent | WT_DHFR_pos139_fwd | This work | Mutagenic PCR primer | NNSTTCCACGATGCTGATGCG |
| Sequence-based reagent | WT_DHFR_pos140_fwd | This work | Mutagenic PCR primer | NNSCACGATGCTGATGCGCAG |

*Appendix 1—key resources table continued*

| Reagent type (species) or resource | Designation | Source or reference | Identifiers | Additional information |
|---|---|---|---|---|
| Sequence-based re-agent | WT_DHFR_pos140_fwd2 | This work | Mutagenic PCR primer | NNSCACGATGCTGATGCGCAG |
| Sequence-based re-agent | WT_DHFR_pos141_fwd | This work | Mutagenic PCR primer | NNSGATGCTGATGCGCAGAACT |
| Sequence-based re-agent | WT_DHFR_pos142_fwd | This work | Mutagenic PCR primer | NNSGCTGATGCGCAGAACTCTC |
| Sequence-based re-agent | WT_DHFR_pos143_fwd | This work | Mutagenic PCR primer | NNSGATGCGCAGAACTCTCA-CAG |
| Sequence-based re-agent | WT_DHFR_pos144_fwd | This work | Mutagenic PCR primer | NNSGCGCAGAACTCTCACAGC |
| Sequence-based re-agent | WT_DHFR_pos145_fwd | This work | Mutagenic PCR primer | NNSCAGAACTCTCACAGCTATTGCTTTG |
| Sequence-based re-agent | WT_DHFR_pos146_fwd | This work | Mutagenic PCR primer | NNSAACTCTCACAGCTATTGCTTTGAGATT |
| Sequence-based re-agent | WT_DHFR_pos147_fwd | This work | Mutagenic PCR primer | NNSTCTCACAGCTATTGCTTTGAGATTCT |
| Sequence-based re-agent | WT_DHFR_pos148_fwd | This work | Mutagenic PCR primer | NNSCACAGCTATTGCTTTGAGATTCTGG |
| Sequence-based re-agent | WT_DHFR_pos149_fwd | This work | Mutagenic PCR primer | NNSAGCTATTGCTTTGAGATTCTGGAG |
| Sequence-based re-agent | WT_DHFR_pos150_fwd | This work | Mutagenic PCR primer | NNSTATTGCTTTGAGATTCTGGAGCG |
| Sequence-based re-agent | WT_DHFR_pos151_fwd | This work | Mutagenic PCR primer | NNSTGCTTTGAGATTCTGGAGCG |
| Sequence-based re-agent | WT_DHFR_pos152_fwd | This work | Mutagenic PCR primer | NNSTTTGAGATTCTGGAGCGGC |
| Sequence-based re-agent | WT_DHFR_pos153_fwd | This work | Mutagenic PCR primer | NNSGAGATTCTGGAGCGGCGG |
| Sequence-based re-agent | WT_DHFR_pos154_fwd | This work | Mutagenic PCR primer | NNSATTCTGGAGCGGCGGTAA |
| Sequence-based re-agent | WT_DHFR_pos155_fwd | This work | Mutagenic PCR primer | NNSCTGGAGCGGCGGTAACAG |
| Sequence-based re-agent | WT_DHFR_pos156_fwd | This work | Mutagenic PCR primer | NNSGAGCGGCGGTAACAGGCG |
| Sequence-based re-agent | WT_DHFR_pos157_fwd | This work | Mutagenic PCR primer | NNSCGGCGGTAACAGGCGTCG |

*Appendix 1—key resources table continued*

| Reagent type (species) or resource | Designation | Source or reference | Identifiers | Additional information |
|---|---|---|---|---|
| Sequence-based re-agent | WT_DHFR_pos158_fwd | This work | Mutagenic PCR primer | NNSCGGTAACAGGCGTCGACA |
| Sequence-based re-agent | WT_DHFR_pos159_fwd | This work | Mutagenic PCR primer | NNSTAACAGGCGTCGACAAGCT |
| Sequence-based re-agent | WT_DHFR_pos2_rev | This work | Mutagenic PCR primer | CATGGTATATCTCCTTATTAAAGTTAAA |
| Sequence-based re-agent | WT_DHFR_pos2_rev2 | This work | Mutagenic PCR primer | CATGGTATATCTCATTATTAAAGT TAAAC |
| Sequence-based re-agent | WT_DHFR_pos3_rev | This work | Mutagenic PCR primer | GATCATGGTATATCTCCTTATTA AAGTT |
| Sequence-based re-agent | WT_DHFR_pos4_rev | This work | Mutagenic PCR primer | ACTGATCATGGTATATCTCCTT ATTAAA |
| Sequence-based re-agent | WT_DHFR_pos5_rev | This work | Mutagenic PCR primer | CAGACTGATCATGGTATATCTC CTTATT |
| Sequence-based re-agent | WT_DHFR_pos6_rev | This work | Mutagenic PCR primer | AATCAGACTGATCATGGTATAT CTCCTT |
| Sequence-based re-agent | WT_DHFR_pos7_rev | This work | Mutagenic PCR primer | CGCAATCAGACTGATCATGGT ATATCT |
| Sequence-based re-agent | WT_DHFR_pos8_rev | This work | Mutagenic PCR primer | CGCCGCAATCAGACTGATC |
| Sequence-based re-agent | WT_DHFR_pos8_rev2 | This work | Mutagenic PCR primer | CGCCGCAATCAGACTGATC |
| Sequence-based re-agent | WT_DHFR_pos9_rev | This work | Mutagenic PCR primer | TAACGCCGCAATCAGACTGA |
| Sequence-based re-agent | WT_DHFR_pos10_rev | This work | Mutagenic PCR primer | CGCTAACGCCGCAATCAG |
| Sequence-based re-agent | WT_DHFR_pos11_rev | This work | Mutagenic PCR primer | TACCGCTAACGCCGCAAT |
| Sequence-based re-agent | WT_DHFR_pos12_rev | This work | Mutagenic PCR primer | ATCTACCGCTAACGCCGC |
| Sequence-based re-agent | WT_DHFR_pos13_rev | This work | Mutagenic PCR primer | GCGATCTACCGCTAACGC |
| Sequence-based re-agent | WT_DHFR_pos14_rev | This work | Mutagenic PCR primer | AACGCGATCTACCGCTAAC |
| Sequence-based re-agent | WT_DHFR_pos15_rev | This work | Mutagenic PCR primer | GATAACGCGATCTACCGCTAAC |

*Appendix 1—key resources table continued*

| Reagent type (species) or resource | Designation | Source or reference | Identifiers | Additional information |
|---|---|---|---|---|
| Sequence-based re-agent | WT_DHFR_pos16_rev | This work | Mutagenic PCR primer | GCCGATAACGCGATCTACC |
| Sequence-based re-agent | WT_DHFR_pos17_rev | This work | Mutagenic PCR primer | CATGCCGATAACGCGATCTAC |
| Sequence-based re-agent | WT_DHFR_pos18_rev | This work | Mutagenic PCR primer | TTCCATGCCGATAACGCG |
| Sequence-based re-agent | WT_DHFR_pos19_rev | This work | Mutagenic PCR primer | GTTTTCCATGCCGATAACGC |
| Sequence-based re-agent | WT_DHFR_pos20_rev | This work | Mutagenic PCR primer | GGCGTTTTCCATGCCGATAACG |
| Sequence-based re-agent | WT_DHFR_pos21_rev | This work | Mutagenic PCR primer | CATGGCGTTTTCCATGCC |
| Sequence-based re-agent | WT_DHFR_pos22_rev | This work | Mutagenic PCR primer | CGGCATGGCGTTTTCCAT |
| Sequence-based re-agent | WT_DHFR_pos22_rev2 | This work | Mutagenic PCR primer | CGGCATGGCGTTTTCCATG |
| Sequence-based re-agent | WT_DHFR_pos23_rev | This work | Mutagenic PCR primer | CCACGGCATGGCGTTTTC |
| Sequence-based re-agent | WT_DHFR_pos24_rev | This work | Mutagenic PCR primer | GTTCCACGGCATGGCGTT |
| Sequence-based re-agent | WT_DHFR_pos25_rev | This work | Mutagenic PCR primer | CAGGTTCCACGGCATGGC |
| Sequence-based re-agent | WT_DHFR_pos26_rev | This work | Mutagenic PCR primer | AGGCAGGTTCCACGGCAT |
| Sequence-based re-agent | WT_DHFR_pos27_rev | This work | Mutagenic PCR primer | GGCAGGCAGGTTCCACGG |
| Sequence-based re-agent | WT_DHFR_pos28_rev | This work | Mutagenic PCR primer | ATCGGCAGGCAGGTTCCA |
| Sequence-based re-agent | WT_DHFR_pos29_rev | This work | Mutagenic PCR primer | GAGATCGGCAGGCAGGTT |
| Sequence-based re-agent | WT_DHFR_pos30_rev | This work | Mutagenic PCR primer | GGCGAGATCGGCAGGCAG |
| Sequence-based re-agent | WT_DHFR_pos31_rev | This work | Mutagenic PCR primer | CCAGGCGAGATCGGCAGG |
| Sequence-based re-agent | WT_DHFR_pos32_rev | This work | Mutagenic PCR primer | AAACCAGGCGAGATCGGC |

*Appendix 1—key resources table continued*

| Reagent type (species) or resource | Designation | Source or reference | Identifiers | Additional information |
|---|---|---|---|---|
| Sequence-based reagent | WT_DHFR_pos33_rev | This work | Mutagenic PCR primer | TTTAAACCAGGCGAGATCGG |
| Sequence-based reagent | WT_DHFR_pos34_rev | This work | Mutagenic PCR primer | GCGTTTAAACCAGGCGAGAT |
| Sequence-based reagent | WT_DHFR_pos35_rev | This work | Mutagenic PCR primer | GTTGCGTTTAAACCAGGCGA |
| Sequence-based reagent | WT_DHFR_pos36_rev | This work | Mutagenic PCR primer | GGTGTTGCGTTTAAACCAGG |
| Sequence-based reagent | WT_DHFR_pos37_rev | This work | Mutagenic PCR primer | TAAGGTGTTGCGTTTAAAC-CAGG |
| Sequence-based reagent | WT_DHFR_pos38_rev | This work | Mutagenic PCR primer | ATTTAAGGTGTTGCGTTTAAAC-CAGG |
| Sequence-based reagent | WT_DHFR_pos39_rev | This work | Mutagenic PCR primer | TTTATTTAAGGTGTTGCG TTTAAACCAG |
| Sequence-based reagent | WT_DHFR_pos40_rev | This work | Mutagenic PCR primer | GGGTTTATTTAAGGTGTTGCG TTTAAAC |
| Sequence-based reagent | WT_DHFR_pos41_rev | This work | Mutagenic PCR primer | CACGGGTTTATTTAAGGTG TTGCGT |
| Sequence-based reagent | WT_DHFR_pos42_rev | This work | Mutagenic PCR primer | AATCACGGGTTTATTTAAGGTG TTGC |
| Sequence-based reagent | WT_DHFR_pos42_rev2 | This work | Mutagenic PCR primer | AATCACGGGTTTATTTAAGGTG TTGC |
| Sequence-based reagent | WT_DHFR_pos43_rev | This work | Mutagenic PCR primer | CATAATCACGGGTTTA TTTAAGGTGTTG |
| Sequence-based reagent | WT_DHFR_pos43_rev2 | This work | Mutagenic PCR primer | CATAATCACGGGTTTA TTTAAGGTGTTG |
| Sequence-based reagent | WT_DHFR_pos44_rev | This work | Mutagenic PCR primer | GCCCATAATCACGGGTTTA TTTAAGG |
| Sequence-based reagent | WT_DHFR_pos45_rev | This work | Mutagenic PCR primer | GCGGCCCATAATCACGGG |
| Sequence-based reagent | WT_DHFR_pos46_rev | This work | Mutagenic PCR primer | ATGGCGGCCCATAATCAC |
| Sequence-based reagent | WT_DHFR_pos47_rev | This work | Mutagenic PCR primer | GGTATGGCGGCCCATAATC |
| Sequence-based reagent | WT_DHFR_pos48_rev | This work | Mutagenic PCR primer | CCAGGTATGGCGGCCCATA |

*Appendix 1—key resources table continued*

| Reagent type (species) or resource | Designation | Source or reference | Identifiers | Additional information |
|---|---|---|---|---|
| Sequence-based reagent | WT_DHFR_pos49_rev | This work | Mutagenic PCR primer | TTCCCAGGTATGGCGGCC |
| Sequence-based reagent | WT_DHFR_pos50_rev | This work | Mutagenic PCR primer | TGATTCCCAGGTATGGCGGC |
| Sequence-based reagent | WT_DHFR_pos51_rev | This work | Mutagenic PCR primer | GATTGATTCCCAGGTATGGCGG |
| Sequence-based reagent | WT_DHFR_pos52_rev | This work | Mutagenic PCR primer | ACCGATTGATTCCCAGGTATG |
| Sequence-based reagent | WT_DHFR_pos53_rev | This work | Mutagenic PCR primer | ACGACCGATTGATTCCCAG |
| Sequence-based reagent | WT_DHFR_pos54_rev | This work | Mutagenic PCR primer | CGGACGACCGATTGATTCC |
| Sequence-based reagent | WT_DHFR_pos55_rev | This work | Mutagenic PCR primer | CAACGGACGACCGATTGATTC |
| Sequence-based reagent | WT_DHFR_pos56_rev | This work | Mutagenic PCR primer | TGGCAACGGACGACCGAT |
| Sequence-based reagent | WT_DHFR_pos57_rev | This work | Mutagenic PCR primer | TCCTGGCAACGGACGACC |
| Sequence-based reagent | WT_DHFR_pos58_rev | This work | Mutagenic PCR primer | GCGTCCTGGCAACGGACG |
| Sequence-based reagent | WT_DHFR_pos59_rev | This work | Mutagenic PCR primer | TTTGCGTCCTGGCAACGG |
| Sequence-based reagent | WT_DHFR_pos60_rev | This work | Mutagenic PCR primer | ATTTTTGCGTCCTGGCAAC |
| Sequence-based reagent | WT_DHFR_pos61_rev | This work | Mutagenic PCR primer | AATATTTTTGCGTCCTGGCAAC |
| Sequence-based reagent | WT_DHFR_pos62_rev | This work | Mutagenic PCR primer | GATAATATTTTTGCGTCC TGGCAAC |
| Sequence-based reagent | WT_DHFR_pos63_rev | This work | Mutagenic PCR primer | GAGGATAATATTTTTGCGTCC TGGC |
| Sequence-based reagent | WT_DHFR_pos64_rev | This work | Mutagenic PCR primer | GCTGAGGATAATATTTTTGCG TCCTG |
| Sequence-based reagent | WT_DHFR_pos65_rev | This work | Mutagenic PCR primer | ACTGCTGAGGATAATA TTTTTGCGTCCT |
| Sequence-based reagent | WT_DHFR_pos66_rev | This work | Mutagenic PCR primer | TTGACTGCTGAGGATAATA TTTTTGCG |

*Appendix 1—key resources table continued*

| Reagent type (species) or resource | Designation | Source or reference | Identifiers | Additional information |
|---|---|---|---|---|
| Sequence-based reagent | WT_DHFR_pos66_rev2 | This work | Mutagenic PCR primer | TTGACTGCTGAGGATAATA TTTTTGC |
| Sequence-based reagent | WT_DHFR_pos67_rev | This work | Mutagenic PCR primer | CGGTTGACTGCTGAGGATAATA TTTTTG |
| Sequence-based reagent | WT_DHFR_pos67_rev2 | This work | Mutagenic PCR primer | CGGTTGACTGCTGAGGATAATA TTTTTG |
| Sequence-based reagent | WT_DHFR_pos68_rev | This work | Mutagenic PCR primer | ACCCGGTTGACTGCTGAG |
| Sequence-based reagent | WT_DHFR_pos69_rev | This work | Mutagenic PCR primer | CGTACCCGGTTGACTGCT |
| Sequence-based reagent | WT_DHFR_pos70_rev | This work | Mutagenic PCR primer | GTCCGTACCCGGTTGACT |
| Sequence-based reagent | WT_DHFR_pos71_rev | This work | Mutagenic PCR primer | ATCGTCCGTACCCGGTTG |
| Sequence-based reagent | WT_DHFR_pos72_rev | This work | Mutagenic PCR primer | GCGATCGTCCGTACCCGG |
| Sequence-based reagent | WT_DHFR_pos73_rev | This work | Mutagenic PCR primer | TACGCGATCGTCCGTACC |
| Sequence-based reagent | WT_DHFR_pos73_rev2 | This work | Mutagenic PCR primer | TACGCGATCGTCCGTACC |
| Sequence-based reagent | WT_DHFR_pos74_rev | This work | Mutagenic PCR primer | CGTTACGCGATCGTCCGT |
| Sequence-based reagent | WT_DHFR_pos74_rev2 | This work | Mutagenic PCR primer | CGTTACGCGATCGTCCGTAC |
| Sequence-based reagent | WT_DHFR_pos75_rev | This work | Mutagenic PCR primer | CCACGTTACGCGATCGTC |
| Sequence-based reagent | WT_DHFR_pos76_rev | This work | Mutagenic PCR primer | CACCCACGTTACGCGATC |
| Sequence-based reagent | WT_DHFR_pos77_rev | This work | Mutagenic PCR primer | CTTCACCCACGTTACGCG |
| Sequence-based reagent | WT_DHFR_pos78_rev | This work | Mutagenic PCR primer | CGACTTCACCCACGTTACG |
| Sequence-based reagent | WT_DHFR_pos79_rev | This work | Mutagenic PCR primer | CACCGACTTCACCCACGTTAC |
| Sequence-based reagent | WT_DHFR_pos80_rev | This work | Mutagenic PCR primer | ATCCACCGACTTCACCCACG TTAC |

*Appendix 1—key resources table continued*

| Reagent type (species) or resource | Designation | Source or reference | Identifiers | Additional information |
|---|---|---|---|---|
| Sequence-based reagent | WT_DHFR_pos80_rev2 | This work | Mutagenic PCR primer | ATCCACCGACTTCACCCAC |
| Sequence-based reagent | WT_DHFR_pos81_rev | This work | Mutagenic PCR primer | TTCATCCACCGACTTCACCCA |
| Sequence-based reagent | WT_DHFR_pos82_rev | This work | Mutagenic PCR primer | GGCTTCATCCACCGACTTCAC |
| Sequence-based reagent | WT_DHFR_pos82_rev2 | This work | Mutagenic PCR primer | GGCTTCATCCACCGACTTCAC |
| Sequence-based reagent | WT_DHFR_pos83_rev | This work | Mutagenic PCR primer | GATGGCTTCATCCACCGAC |
| Sequence-based reagent | WT_DHFR_pos84_rev | This work | Mutagenic PCR primer | CGCGATGGCTTCATCCAC |
| Sequence-based reagent | WT_DHFR_pos84_rev2 | This work | Mutagenic PCR primer | CGCGATGGCTTCATCCAC |
| Sequence-based reagent | WT_DHFR_pos85_rev | This work | Mutagenic PCR primer | CGCCGCGATGGCTTCATC |
| Sequence-based reagent | WT_DHFR_pos86_rev | This work | Mutagenic PCR primer | ACACGCCGCGATGGCTTC |
| Sequence-based reagent | WT_DHFR_pos87_rev | This work | Mutagenic PCR primer | ACCACACGCCGCGATGGC |
| Sequence-based reagent | WT_DHFR_pos88_rev | This work | Mutagenic PCR primer | GTCACCACACGCCGCGAT |
| Sequence-based reagent | WT_DHFR_pos89_rev | This work | Mutagenic PCR primer | TACGTCACCACACGCCGC |
| Sequence-based reagent | WT_DHFR_pos89_rev2 | This work | Mutagenic PCR primer | TACGTCACCACACGCCG |
| Sequence-based reagent | WT_DHFR_pos90_rev | This work | Mutagenic PCR primer | TGGTACGTCACCACACGC |
| Sequence-based reagent | WT_DHFR_pos91_rev | This work | Mutagenic PCR primer | TTCTGGTACGTCACCACACGC |
| Sequence-based reagent | WT_DHFR_pos92_rev | This work | Mutagenic PCR primer | GATTCTGGTACGTCACCA-CACG |
| Sequence-based reagent | WT_DHFR_pos93_rev | This work | Mutagenic PCR primer | CATGATTTCTGGTACGTCACCA-CAC |
| Sequence-based reagent | WT_DHFR_pos94_rev | This work | Mutagenic PCR primer | CACCATGATTTCTGGTACG TCACC |

*Appendix 1—key resources table continued*

| Reagent type (species) or resource | Designation | Source or reference | Identifiers | Additional information |
|---|---|---|---|---|
| Sequence-based re-agent | WT_DHFR_pos95_rev | This work | Mutagenic PCR primer | AATCACCATGATTTCTGGTACGTCA |
| Sequence-based re-agent | WT_DHFR_pos95_rev2 | This work | Mutagenic PCR primer | AATCACCATGATTTCTGGTACGTC |
| Sequence-based re-agent | WT_DHFR_pos96_rev | This work | Mutagenic PCR primer | GCCAATCACCATGATTTCTGGTAC |
| Sequence-based re-agent | WT_DHFR_pos97_rev | This work | Mutagenic PCR primer | GCCGCCAATCACCATGATTT |
| Sequence-based re-agent | WT_DHFR_pos98_rev | This work | Mutagenic PCR primer | ACCGCCGCCAATCACCATGATTTC |
| Sequence-based re-agent | WT_DHFR_pos99_rev | This work | Mutagenic PCR primer | GCGACCGCCGCCAATCAC |
| Sequence-based re-agent | WT_DHFR_pos100_rev | This work | Mutagenic PCR primer | AACGCGACCGCCGCCAAT |
| Sequence-based re-agent | WT_DHFR_pos101_rev | This work | Mutagenic PCR primer | ATAAACGCGACCGCCGCC |
| Sequence-based re-agent | WT_DHFR_pos102_rev | This work | Mutagenic PCR primer | TTCATAAACGCGACCGCC |
| Sequence-based re-agent | WT_DHFR_pos103_rev | This work | Mutagenic PCR primer | CTGTTCATAAACGCGACCG |
| Sequence-based re-agent | WT_DHFR_pos104_rev | This work | Mutagenic PCR primer | GAACTGTTCATAAACGCGACC |
| Sequence-based re-agent | WT_DHFR_pos104_rev2 | This work | Mutagenic PCR primer | GAACTGTTCATAAACGCGACCG |
| Sequence-based re-agent | WT_DHFR_pos105_rev | This work | Mutagenic PCR primer | CAAGAACTGTTCATAAACGC-GAC |
| Sequence-based re-agent | WT_DHFR_pos106_rev | This work | Mutagenic PCR primer | TGGCAAGAACTGTTCATAAACGC |
| Sequence-based re-agent | WT_DHFR_pos107_rev | This work | Mutagenic PCR primer | TTTTGGCAAGAACTGTTCATAAACG |
| Sequence-based re-agent | WT_DHFR_pos107_rev2 | This work | Mutagenic PCR primer | TTTTGGCAAGAACTGTTCATAAACG |
| Sequence-based re-agent | WT_DHFR_pos108_rev | This work | Mutagenic PCR primer | CGCTTTTGGCAAGAACTGTTCATAAA |
| Sequence-based re-agent | WT_DHFR_pos109_rev | This work | Mutagenic PCR primer | TTGCGCTTTTGGCAAGAACT |

*Appendix 1—key resources table continued*

| Reagent type (species) or resource | Designation | Source or reference | Identifiers | Additional information |
|---|---|---|---|---|
| Sequence-based re-agent | WT_DHFR_pos110_rev | This work | Mutagenic PCR primer | TTTTTGCGCTTTTGGCAAGAAC |
| Sequence-based re-agent | WT_DHFR_pos111_rev | This work | Mutagenic PCR primer | CAGTTTTTGCGCTTTTGGCAAG |
| Sequence-based re-agent | WT_DHFR_pos112_rev | This work | Mutagenic PCR primer | ATACAGTTTTTGCGC TTTTGGCAA |
| Sequence-based re-agent | WT_DHFR_pos113_rev | This work | Mutagenic PCR primer | CAGATACAGTTTTTGCGC TTTTGG |
| Sequence-based re-agent | WT_DHFR_pos114_rev | This work | Mutagenic PCR primer | CGTCAGATACAGTTTTTGCGC TTTT |
| Sequence-based re-agent | WT_DHFR_pos115_rev | This work | Mutagenic PCR primer | ATGCGTCAGATACAG TTTTTGCG |
| Sequence-based re-agent | WT_DHFR_pos116_rev | This work | Mutagenic PCR primer | GATATGCGTCAGATACAGTT TTTGCG |
| Sequence-based re-agent | WT_DHFR_pos117_rev | This work | Mutagenic PCR primer | GTCGATATGCGTCAGATACAG TTTTTG |
| Sequence-based re-agent | WT_DHFR_pos118_rev | This work | Mutagenic PCR primer | TGCGTCGATATGCGTCAGATA |
| Sequence-based re-agent | WT_DHFR_pos118_rev2 | This work | Mutagenic PCR primer | TGCGTCGATATGCGTCAGATAC |
| Sequence-based re-agent | WT_DHFR_pos119_rev | This work | Mutagenic PCR primer | TTCTGCGTCGATATGCGTCA |
| Sequence-based re-agent | WT_DHFR_pos120_rev | This work | Mutagenic PCR primer | CACTTCTGCGTCGATATGCG |
| Sequence-based re-agent | WT_DHFR_pos121_rev | This work | Mutagenic PCR primer | TTCCACTTCTGCGTCGATATG |
| Sequence-based re-agent | WT_DHFR_pos122_rev | This work | Mutagenic PCR primer | GCCTTCCACTTCTGCGTC |
| Sequence-based re-agent | WT_DHFR_pos123_rev | This work | Mutagenic PCR primer | GTCGCCTTCCACTTCTGC |
| Sequence-based re-agent | WT_DHFR_pos124_rev | This work | Mutagenic PCR primer | GGTGTCGCCTTCCACTTC |
| Sequence-based re-agent | WT_DHFR_pos125_rev | This work | Mutagenic PCR primer | ATGGGTGTCGCCTTCCAC |
| Sequence-based re-agent | WT_DHFR_pos126_rev | This work | Mutagenic PCR primer | GAAATGGGTGTCGCCTTCC |

*Appendix 1—key resources table continued*

| Reagent type (species) or resource | Designation | Source or reference | Identifiers | Additional information |
|---|---|---|---|---|
| Sequence-based reagent | WT_DHFR_pos127_rev | This work | Mutagenic PCR primer | CGGGAAATGGGTGTCGCC |
| Sequence-based reagent | WT_DHFR_pos128_rev | This work | Mutagenic PCR primer | ATCCGGGAAATGGGTGTC |
| Sequence-based reagent | WT_DHFR_pos129_rev | This work | Mutagenic PCR primer | GTAATCCGGGAAATGGGTGTC |
| Sequence-based reagent | WT_DHFR_pos130_rev | This work | Mutagenic PCR primer | CTCGTAATCCGGGAAATGGG |
| Sequence-based reagent | WT_DHFR_pos131_rev | This work | Mutagenic PCR primer | CGGCTCGTAATCCGGGAA |
| Sequence-based reagent | WT_DHFR_pos131_rev2 | This work | Mutagenic PCR primer | CGGCTCGTAATCCGGGAAATG |
| Sequence-based reagent | WT_DHFR_pos132_rev | This work | Mutagenic PCR primer | ATCCGGCTCGTAATCCGG |
| Sequence-based reagent | WT_DHFR_pos133_rev | This work | Mutagenic PCR primer | GTCATCCGGCTCGTAATCC |
| Sequence-based reagent | WT_DHFR_pos134_rev | This work | Mutagenic PCR primer | CCAGTCATCCGGCTCGTA |
| Sequence-based reagent | WT_DHFR_pos135_rev | This work | Mutagenic PCR primer | TTCCCAGTCATCCGGCTC |
| Sequence-based reagent | WT_DHFR_pos135_rev2 | This work | Mutagenic PCR primer | TTCCCAGTCATCCGGCTC |
| Sequence-based reagent | WT_DHFR_pos136_rev | This work | Mutagenic PCR primer | CGATTCCCAGTCATCCGG |
| Sequence-based reagent | WT_DHFR_pos136_rev2 | This work | Mutagenic PCR primer | CGATTCCCAGTCATCCGGC |
| Sequence-based reagent | WT_DHFR_pos137_rev | This work | Mutagenic PCR primer | TACCGATTCCCAGTCATCCG |
| Sequence-based reagent | WT_DHFR_pos138_rev | This work | Mutagenic PCR primer | GAATACCGATTCCCAGTCATCC |
| Sequence-based reagent | WT_DHFR_pos139_rev | This work | Mutagenic PCR primer | GCTGAATACCGATTCCCAGTC |
| Sequence-based reagent | WT_DHFR_pos140_rev | This work | Mutagenic PCR primer | TTCGCTGAATACCGATTCCCA |
| Sequence-based reagent | WT_DHFR_pos140_rev2 | This work | Mutagenic PCR primer | TTCGCTGAATACCGATTCCCAG |

*Appendix 1—key resources table continued*

| Reagent type (species) or resource | Designation | Source or reference | Identifiers | Additional information |
|---|---|---|---|---|
| Sequence-based reagent | WT_DHFR_pos141_rev | This work | Mutagenic PCR primer | GAATTCGCTGAATACCGATTCCC |
| Sequence-based reagent | WT_DHFR_pos142_rev | This work | Mutagenic PCR primer | GTGGAATTCGCTGAATACCGATTC |
| Sequence-based reagent | WT_DHFR_pos143_rev | This work | Mutagenic PCR primer | ATCGTGGAATTCGCTGAATACC |
| Sequence-based reagent | WT_DHFR_pos144_rev | This work | Mutagenic PCR primer | AGCATCGTGGAATTCGCTG |
| Sequence-based reagent | WT_DHFR_pos145_rev | This work | Mutagenic PCR primer | ATCAGCATCGTGGAATTCGC |
| Sequence-based reagent | WT_DHFR_pos146_rev | This work | Mutagenic PCR primer | CGCATCAGCATCGTGGAATT |
| Sequence-based reagent | WT_DHFR_pos147_rev | This work | Mutagenic PCR primer | CTGCGCATCAGCATCGTG |
| Sequence-based reagent | WT_DHFR_pos148_rev | This work | Mutagenic PCR primer | GTTCTGCGCATCAGCATC |
| Sequence-based reagent | WT_DHFR_pos149_rev | This work | Mutagenic PCR primer | AGAGTTCTGCGCATCAGC |
| Sequence-based reagent | WT_DHFR_pos150_rev | This work | Mutagenic PCR primer | GTGAGAGTTCTGCGCATCAG |
| Sequence-based reagent | WT_DHFR_pos151_rev | This work | Mutagenic PCR primer | GCTGTGAGAGTTCTGCGC |
| Sequence-based reagent | WT_DHFR_pos152_rev | This work | Mutagenic PCR primer | ATAGCTGTGAGAGTTCTGCG |
| Sequence-based reagent | WT_DHFR_pos153_rev | This work | Mutagenic PCR primer | GCAATAGCTGTGAGAGTTCTGC |
| Sequence-based reagent | WT_DHFR_pos154_rev | This work | Mutagenic PCR primer | AAAGCAATAGCTGTGAGAGTTCTG |
| Sequence-based reagent | WT_DHFR_pos155_rev | This work | Mutagenic PCR primer | CTCAAAGCAATAGCTGTGAGAGTTC |
| Sequence-based reagent | WT_DHFR_pos156_rev | This work | Mutagenic PCR primer | AATCTCAAAGCAATAGCTGTGAGAGTTC |
| Sequence-based reagent | WT_DHFR_pos157_rev | This work | Mutagenic PCR primer | CAGAATCTCAAAGCAATAGCTGTGAGAG |
| Sequence-based reagent | WT_DHFR_pos158_rev | This work | Mutagenic PCR primer | CTCCAGAATCTCAAAGCAATAGCTG |

*Appendix 1—key resources table continued*

| Reagent type (species) or resource | Designation | Source or reference | Identifiers | Additional information |
|---|---|---|---|---|
| Sequence-based re-agent | WT_DHFR_pos159_rev | This work | Mutagenic PCR primer | CCGCTCCAGAATCTCAAAGC |
| Sequence-based re-agent | SL1_FWD | This work | Round one amplicon PCR primer | CACTCTTTCCCTACACGACGCGTCTTCCGATCTNNNNACTTTAATAACGAGATATACCATG |
| Sequence-based re-agent | SL1_REV | This work | Round one amplicon PCR primer | TGACTGGAGTTCAGACGTGTGCTCTTCCGATCTNNNNGTATGGCGGCCCATAAT |
| Sequence-based re-agent | SL2_FWD | This work | Round one amplicon PCR primer | CACTCTTTCCCTACACGACGCGTCTTCCGATCTNNNNACACCTTAAATAAACCCGTG |
| Sequence-based re-agent | SL2_REV | This work | Round one amplicon PCR primer | TGACTGGAGTTCAGACGTGTGCTCTTCCGATCTNNNNCACGCCGCGATGGC |
| Sequence-based re-agent | SL3_FWD | This work | Round one amplicon PCR primer | CACTCTTTCCCTACACGACGCGTCTTCCGATCTNNNNTGAAGTCGGTGGATGAA |
| Sequence-based re-agent | SL3_REV | This work | Round one amplicon PCR primer | TGACTGGAGTTCAGACGTGTGCTCTTCCGATCTNNNNGAAATGGGTGTCGCC |
| Sequence-based re-agent | SL4_FWD | This work | Round one amplicon PCR primer | CACTCTTTCCCTACACGACGCTCTTCCGATCTNNNNCGACGCAGAAGTGGAA |
| Sequence-based re-agent | SL4_REV | This work | Round one amplicon PCR primer | TGACTGGAGTTCAGACGTGTGCTCTTCCGATCTNNNNGCTTGTCGACGCCTG |
| Sequence-based re-agent | D501 | Illumina/ *Reynolds et al., 2011* | Round two amplicon PCR primer | AATGATACGGCGACCACCGAGATCTACACTATAGCCTACACTCTTTCCCTACACGAC |
| Sequence-based re-agent | D502 | Illumina/Reynolds et al. Cell 2011 | Round two amplicon PCR primer | AATGATACGGCGACCACCGAGATCTACACATAGAGGCACACTCTTTCCCTACACGAC |
| Sequence-based re-agent | D503 | Illumina/ *Reynolds et al., 2011* | Round two amplicon PCR primer | AATGATACGGCGACCACCGA-GATCTACACCCTATCCTACACTCTTTCCCTACACGAC |
| Sequence-based re-agent | D504 | Illumina/Reynolds et al. Cell 2011 | Round two amplicon PCR primer | AATGATACGGCGACCACCGA-GATCTACACGGCTCTGAACACTCTTTCCTACACGAC |
| Sequence-based re-agent | D505 | Illumina/Reynolds et al. Cell 2011 | Round two amplicon PCR primer | AATGATACGGCGACCACCGA-GATCTACACAGGCGAAGACACTCTTTCCCTACACGAC |
| Sequence-based re-agent | D506 | Illumina/ *Reynolds et al., 2011* | Round two amplicon PCR primer | AATGATACGGCGACCACCGA-GATCTACACTAATCTTAACACTCTTTCCCTACACGAC |
| Sequence-based re-agent | D507 | Illumina/ *Reynolds et al., 2011* | Round two amplicon PCR primer | AATGATACGGCGACCACCGAGATCTACACCAGGACGTACACTCTTTCCCTACACGAC |

*Appendix 1—key resources table continued*

| Reagent type (species) or resource | Designation | Source or reference | Identifiers | Additional information |
|---|---|---|---|---|
| Sequence-based reagent | D508 | Illumina/*Reynolds et al., 2011* | Round two amplicon PCR primer | AATGATACGGCGACCACCGA-GATC TACAC<u>GTACTGAC</u>ACACTC TTTCC CTACACGAC |
| Sequence-based reagent | D701 | Illumina/Reynolds et al. Cell 2011 | Round two amplicon PCR primer | CAAGCAGAAGACGGCATACGA GATC<u>GAGTAATG</u>TGACTGGA GTTCAGACGTG |
| Sequence-based reagent | D702 | Illumina/*Reynolds et al., 2011* | Round two amplicon PCR primer | CAAGCAGAAGACGGCATACGA GATT<u>CTCCGGA</u>GTGACTGGA GTTCAGACGTG |
| Sequence-based reagent | D703 | Illumina/*Reynolds et al., 2011* | Round two amplicon PCR primer | CAAGCAGAAGACGGCATA CGAGAT<u>AATGAGCG</u>GTGA CTGGAGTTCAGACGTG |
| Sequence-based reagent | D704 | Illumina/*Reynolds et al., 2011* | Round two amplicon PCR primer | CAAGCAGAAGACGGCATACG AGAT<u>GGAATCTC</u>GTGACTGG AGTTCAGACGTG |
| Sequence-based reagent | D705 | Illumina/*Reynolds et al., 2011* | Round two amplicon PCR primer | CAAGCAGAAGACGGCATACGA GAT<u>TTCTGAAT</u>GTGACTGGA GTTCAGACGTG |
| Sequence-based reagent | D706 | Illumina/*Reynolds et al., 2011* | Round two amplicon PCR primer | CAAGCAGAAGACGGCATACGA GAT<u>ACGAATTC</u>GTGACTGGAG TTCAGACGTG |
| Sequence-based reagent | D707 | Illumina/*Reynolds et al., 2011* | Round two amplicon PCR primer | CAAGCAGAAGACGGCATACGA GAT<u>AGCTTCAG</u>GTGACTGGAGT TCAGACGTG |
| Sequence-based reagent | D708 | Illumina/*Reynolds et al., 2011* | Round two amplicon PCR primer | CAAGCAGAAGACGGCATAC-GAG AT<u>GCGCATTA</u>GTGACTGGAGTT CAGACGTG |
| Sequence-based reagent | D709 | Illumina/*Reynolds et al., 2011* | Round two amplicon PCR primer | CAAGCAGAAGACGGCATACGA-GAT <u>CATAGCCG</u>GTGACTGGAG TTCAGACGTG |
| Sequence-based reagent | D710 | Illumina/*Reynolds et al., 2011* | Round two amplicon PCR primer | CAAGCAGAAGACGGCATACGA GAT<u>TTCGCGGA</u>GTGACTGGAG TTCAGACGTG |
| Sequence-based reagent | D711 | Illumina/*Reynolds et al., 2011* | Round two amplicon PCR primer | CAAGCAGAAGACGGCATACGA-GA T<u>GCGCGAGA</u>GTGACTGGAGTT CAGACGTG |
| Sequence-based reagent | D712 | Illumina/*Reynolds et al., 2011* | Round two amplicon PCR primer | CAAGCAGAAGACGGCATACGA GAT<u>CTATCGCT</u>GTGACTGGAG TTCAGACGTG |
| Sequence-based reagent | KanSacB_round1_fwd | This work | PCR primer | caggcatctggtgaataa TCCTTTTATGATTTTCTAT CAAACAAAAGAGG |
| Sequence-based reagent | KanSacB_round1_rev | This work | PCR primer | tcaatgcgttcagaacgctca ggattcatGCTTGGTCGGT CATTTCGAAC |
| Sequence-based reagent | KanSacB_round2_fwd/ Anderson_promoter _outer_fwd | This work | PCR primer | gtcaaagcaaaccgttgctgatttatg gcaagccggaagcgcaacaggcat ctggtgaataa |
| Sequence-based reagent | KanSacB_round2_rev/ Anderson_promoter _outer_rev | This work | PCR primer | ccaccacatcgcgcagcggcaatac ggggatttcaatgcgttcagaacgc tcaggattcat |

*Appendix 1—key resources table continued*

| Reagent type (species) or resource | Designation | Source or reference | Identifiers | Additional information |
|---|---|---|---|---|
| Sequence-based re-agent | Anderson_promoter _outer_fwd/KanSacB _round2_fwd | This work | PCR primer | same as KanSacB_round2_fwd /Anderson_promoter_outer_fwd |
| Sequence-based re-agent | Anderson_promoter _inner_fwd | This work | PCR primer | CCTAGGACTGAGCTAGCTG TCAA cgtcagtatatggggatgtttcccc |
| Sequence-based re-agent | Anderson_promoter _inner_rev | This work | PCR primer | GCTAGCTCAGTCCTAGG TATAATGCTAGCAGGAtacctgg cggaaattaaactaagagag |
| Sequence-based re-agent | Anderson_promoter _outer_rev/KanSacB _round2_rev | This work | PCR primer | same as KanSacB_round2_rev /Anderson_promoter_outer_rev |

