## [Decision Letter]

**Acceptance summary:**

The authors combined mutational scanning with structural and biochemical analysis of DHFR against different genetic backgrounds and show how these backgrounds can change the tolerance to mutations. The work provides several important mechanistic insights on the relationship between cellular proteostasis, protein structure and evolution.

**Decision letter after peer review:**

Thank you for submitting your article "Modulating the cellular context broadly reshapes the mutational landscape of a model enzyme" for consideration by *eLife*. Your article has been reviewed by three peer reviewers, including Sarel Jacob Fleishman as the Reviewing Editor, and the evaluation has been overseen by Patricia Wittkopp as the Senior Editor.

The reviewers have discussed the reviews with one another and the Reviewing Editor has drafted this decision to help you prepare a revised submission.

Summary:

Thompson et al. used deep mutational scanning of *E. coli* DHFR to evaluate how the constraints imposed by the cellular environment modulate the mutational tolerance of the enzyme. To this end, selection coefficients of every possible DHFR amino acid substitution were determined in the absence and presence of Lon protease. The authors demonstrate that Lon dramatically transforms the mutational landscape of DHFR. A particularly interesting finding is that Lon largely suppresses the advantageous mutations that, in the absence of Lon, constitute over 23% of all single point mutations. It is suggested that the observed phenomenon can be explained by extensive activity-stability trade-offs, whereby advantageous mutations increase the DHFR activity, but this improvement in activity comes at the expense of reduced thermodynamic stability that renders the mutants sensitive to Lon degradation. The manuscript is clearly written and interesting and nicely adds to our understanding of the relationship between cellular proteostasis and evolution.

Essential revisions:

1) Bacterial fitness depends on the product of the catalytic proficiency (k_cat_/K_M_) and intracellular abundance of an essential enzyme (Dykhuzien, Dean and Hartl, 1987). This dependence was also specifically demonstrated for DHFR in *E. coli* (Bershtein et al., 2015) but isn't discussed in this paper. In the manuscript, the activity of the DHFR mutants is measured as initial velocity at a particular concentration of DHF. However, the comparison between DHFR mutants using this type of measurement is meaningless for mutants that vary substantially in their Km(DHF) values. For example, the reported Km value of L54F is 0.7 μm and that of F31Y is 168 μm – 240 fold higher (Figure 1—source data 3). This means that when the initial velocity for both mutants is measured at 20 μm DHF, the L54F variant operates at a rate close to Vmax, whereas the rate of F31Y is measured way below its Km and, therefore, is far away from its Vmax value. Since the changes in k_cat_ and K_M_ amongst DHFR mutants are not necessarily correlated (e.g., the k_cat_ of variant F31V is close to that of wt but its K_M_ is 2 orders of magnitude higher, Figure 1—source data 3), the differences in the initial velocities at a given concentration of DHF will be sometimes driven by k_cat_ and sometimes by K_M_. The interpretation of these measurements with respect to bacterial fitness is further muddled by the fact that 1) the intracellular concentrations of the mutants are not known, and 2) the intracellular amounts of DHF can rise as a result of low DHFR abundance and/or activity (Kwon et al., 2008), thus affecting the relative importance of Km. Indeed, roughly half of analyzed adaptive mutations appear to have initial velocities lower than that of wt (Figure 4C and Figure 4—figure supplement 3), although the authors claim that the initial velocities are expected to be correlated with selection coefficients (as shown in Figure 1C for a small subset of mutants). Thus, the way the activity of DHFR mutants is measured does not adequately explain the observed distribution of selection coefficients.

For proper interpretation of the selection coefficients, it is therefore important to measure the intracellular abundance of a selection of DHFR mutants on Lon+/- backgrounds and to measure k_cat_ and K_M_ parameters for a subset of advantageous DHFR mutants.

2) Related to point (1) above, the mechanism invoked by the authors to explain why destabilization may increase activity through increased dynamics at the active site is interesting but other mechanisms related to cellular abundance have not been taken into consideration. In particular, DHFR destabilization is known to turn DHFR into a chaperonin client and this interaction may increase cellular levels. As argued in point (1) above, more detailed measurements of cellular abundance and k_cat_,K_M_ determination are needed to produce a consistent interpretation of the results.

3) Results – the authors show that their DMS results are nicely reproducible. However, I don't think that they correlate the DMS results with individually measured selection coefficients (it's not totally clear whether the data shown in Figure 1C is from individual measurements or the DMS). They should do this to establish that the DMS accurately recapitulates individual measurements both in *E. coli* and for purified protein.

4) The selection system is beautifully designed to allow highly sensitive selection conditions, including the identification of better-than-wt DHFR mutants. The experimental conditions in the paper, however, are likely to be different from those that a wild type strain would face. First, the endogenous promoter of folA regulates the DHFR expression via a negative feedback loop: A drop in DHFR activity/abundance results in the upregulation of its expression (Bershtein, et al., 2015). An interesting question is how the distribution of fitness effects of DHFR mutations will be shaped by the presence of such a regulatory expression element. Second, it was demonstrated that the endogenous DHFR levels in *E. coli* strain carrying the chromosomal folA gene are very close to the optimal level, as the increase in activity or abundance of DHFR does not increase fitness (Bhattacharyya et al., 2016). The fact that over 23% of single point DHFR mutations increase bacterial fitness suggests that the intracellular DHFR levels in the selection system are far away from the optimum. Third, there is no difference in the DHFR sequence between naturally occurring *E. coli* B and K-12 strains, even though according to the authors' conclusions, the lack of Lon protease in B strains should have driven the adaptive evolution of DHFR in this strain. It would be helpful if the authors discussed these caveats in the manuscript.

---

## [Author Response]

Essential revisions:1) Bacterial fitness depends on the product of the catalytic proficiency (k_cat_/K_M_) and intracellular abundance of an essential enzyme (Dykhuzien, Dean and Hartl, 1987). This dependence was also specifically demonstrated for DHFR in *E. coli* (Bershtein et al., 2015) but isn't discussed in this paper. In the manuscript, the activity of the DHFR mutants is measured as initial velocity at a particular concentration of DHF. However, the comparison between DHFR mutants using this type of measurement is meaningless for mutants that vary substantially in their Km(DHF) values. For example, the reported Km value of L54F is 0.7 μm and that of F31Y is 168 μm – 240 fold higher (Figure 1—source data 3). This means that when the initial velocity for both mutants is measured at 20 μm DHF, the L54F variant operates at a rate close to Vmax, whereas the rate of F31Y is measured way below its Km and, therefore, is far away from its Vmax value. Since the changes in k_cat_ and K_M_ amongst DHFR mutants are not necessarily correlated (e.g., the k_cat_ of variant F31V is close to that of wt but its K_M_ is 2 orders of magnitude higher, Figure 1—source data 3), the differences in the initial velocities at a given concentration of DHF will be sometimes driven by k_cat_ and sometimes by K_M_. The interpretation of these measurements with respect to bacterial fitness is further muddled by the fact that 1) the intracellular concentrations of the mutants are not known, and 2) the intracellular amounts of DHF can rise as a result of low DHFR abundance and/or activity (Kwon et al., 2008), thus affecting the relative importance of Km. Indeed, roughly half of analyzed adaptive mutations appear to have initial velocities lower than that of wt (Figure 4C and Figure 4—figure supplement 3), although the authors claim that the initial velocities are expected to be correlated with selection coefficients (as shown in Figure 1C for a small subset of mutants). Thus, the way the activity of DHFR mutants is measured does not adequately explain the observed distribution of selection coefficients.For proper interpretation of the selection coefficients, it is therefore important to measure the intracellular abundance of a selection of DHFR mutants on Lon+/- backgrounds and to measure k_cat_ and K_M_ parameters for a subset of advantageous DHFR mutants.

We thank the reviewers for this important comment concerning the key factors that affect the selection pressure in our assay. The phrasing in original submission suggested a straight-forward, single explanation (in vitro catalytic activity) for the selection pressure. While there is a correlation between selection coefficients and the in vitro velocity of the mutants used in the calibration of our selection conditions, this relationship is clearly too simplistic. We agree with the reviewers that differences in selection coefficients arise from an interplay of several factors including DHFR catalytic activity, intracellular DHFR abundance, and substrate concentration (in addition to potential other factors such as changes in feedback regulation and chaperone activity, as also pointed out by the reviewers and discussed further below).

As mentioned by the reviewers, (Bershtein et al., 2015) relate growth rate to abundance • k_cat_/K_M_, but their data (Author response image 1) also illustrate the difficulty with these measurements. The DHFR variants studied by Bershtein exhibit WT-like growth rates that differ by less than a factor of two, and a considerable fraction of the variants appear to be in a plateau region where growth rate is largely independent of changes in abundance • k_cat_/K_M,_ as predicted by a metabolic flux model. However, in the region before the plateau (right inset) where one would expect a dependence of growth rate on DHFR abundance • k_cat_/K_M_, interpretation becomes difficult because of limits in the accuracy with which the small differences in the relevant parameters can be quantified (abundance measurement errors were not given).

**Author response image 1. sa2fig1:** 

In our revised manuscript we now relate our selection coefficients to both DHFR abundance and catalytic activity. In new experiments, we estimated the intracellular DHFR abundance [DHFR] of 21 DHFR variants using a method adapted from previously used assays (Guerrero et al., 2019; Rodrigues et al., 2016). These new data are presented in Figure 4D and Figure 4—figure supplement 4.To quantify activity and compare relative activities of different DHFR mutants, we previously measured the enzyme velocity_[DHF]_ at several DHF concentrations for each DHFR mutant. We would like to note that velocity can be a more informative metric for comparing DHFR activity than k_cat_/K_M_, provided that the in vivo DHFR concentration can be estimated. k_cat_/K_M_ can be misleading as the sole metric for comparing mutants of the same enzyme acting on the same substrate under all conditions because the relative velocities of the mutant enzymes are additionally dependent on the substrate concentration (see for example Figure 2A (Eisenthal, Danson and Hough, 2007): The ratio of relative velocities for two enzyme variants with the same k_cat_/K_M_ ratio can invert with changing substrate concentration). Directly comparing velocities resolves this problem but requires assuming a relevant intracellular concentration of DHF, which we can estimate from existing data. Results from the Rabinowitz lab show that the concentration of reduced folates (DHF and its polyglutamylated derivatives) is in the low tens of µM for exponentially growing *E. coli* in glucose-rich M9 media (Kwon et al., 2008). While the concentration of reduced folates can rise as a result of low DHFR activity (e.g. reaching ~100 µM after the addition of trimethoprim), the reduced folate concentration returns to approximately the starting concentration within the span of an hour. Because this adaptation to reduced DHFR activity is shorter than the time between our first two timepoints from selection (0 hrs and 2 hrs), we estimate the DHF concentration to be in the tens of µM. (We acknowledge that fully resolving this question would require the measurement of [DHF] for each of the mutants characterized here under the turbidostat growth conditions, ideally at multiple times during an 18-hour growth period to determine if folate levels fluctuate over time. These measurements are highly specialized because of the sensitivity of reduced folates and are beyond our capacity without collaborating with experts. In the absence of such mutant-specific data on in vivo DHF concentrations, k_cat_ and K_M_ values would not provide additional information on in vivo DHFR mutant activity.)

With the new data on DHFR intracellular abundance, we can now estimate cellular DHFR activity as [DHFR] • velocity_[DHF]_. This new analysis resolves the discrepancies for the subset of advantageous DHFR mutants that previously could not be explained simply by the velocity data alone since the mutants did not show increased velocities compared to wild-type. However, total cellular DHFR activity increases as the interplay of velocity and DHFR abundance: In general, advantageous mutants with lower velocities than wild-type have increased DHFR abundance (left part of Figure 4D), qualitatively explaining increased fitness over wild-type (Figure 4E and Figure 4—figure supplement 8).

In the revision, we have added the following to the Results section:

“We first confirmed that the selected advantageous mutations indeed had higher cytosolic DHFR activity (the total rate of conversion of DHF to THF) in ER2566 *∆folA/∆thyA* (–Lon) lysates relative to the activity for WT DHFR (Figure 4—figure supplement 2), consistent with the deep mutational scanning results. […] Moreover, when considering both velocity and abundance the expected total cellular DHFR activity ([DHFR] • velocity) is increased compared to wild-type for the majority of advantageous mutants (Figure 4E, Figure 4—figure supplement 6, positions above the dotted line indicate expected cellular activity greater than wild-type).”

2) Related to point (1) above, the mechanism invoked by the authors to explain why destabilization may increase activity through increased dynamics at the active site is interesting but other mechanisms related to cellular abundance have not been taken into consideration. In particular, DHFR destabilization is known to turn DHFR into a chaperonin client and this interaction may increase cellular levels. As argued in point (1) above, more detailed measurements of cellular abundance and k_cat_,K_M_ determination are needed to produce a consistent interpretation of the results.

Please see our response to the related point (1) above, where we now detail a consistent model for the changed selection coefficients based on both cellular abundance and velocity.

We have also added a brief section on other mechanisms, including that DHFR destabilization is known to turn DHFR into a chaperonin client:

“However, the expected total cellular DHFR activity is not a strong quantitative predictor of the advantageous mutants in –Lon selection (Figure 4—figure supplement 7, Figure 4—figure supplement 8). […] Moreover, Lon suppresses advantageous mutations at least in part by reducing their cellular abundance.”

3) Results – The authors show that their DMS results are nicely reproducible. However, I don't think that they correlate the DMS results with individually measured selection coefficients (it's not totally clear whether the data shown in Figure 1C is from individual measurements or the DMS). They should do this to establish that the DMS accurately recapitulates individual measurements both in *E. coli* and for purified protein.

The data from Figure 1C were obtained from DMS measurements and we have now clarified this point. As suggested by the reviewer, we have added new data on individually measured selection coefficients, which correlate well with the DMS data (Figure 1—figure supplement 3B). We changed the manuscript as follows:

Legend for Figure 1:

“C) Selection coefficients from deep mutational scanning as a function of enzymatic velocity for purified DHFR point mutants measured in vitro.”

Main text:

“For a panel of 14 DHFR mutants, we confirmed that the selection coefficients obtained from deep mutational scanning correlated linearly with growth rates measured separately for the individual variants in a plate reader (Figure 1—figure supplement 3B, Figure 1—source data 2), as expected. Furthermore, under our controlled selection conditions we observed a linear relationship between selection coefficient and in vitro velocity (Figure 1C) at cytosolic substrate concentrations(Bennett et al., 2009; Kwon et al., 2008) for these DHFR mutants (Figure 1—source data 3).”

4) The selection system is beautifully designed to allow highly sensitive selection conditions, including the identification of better-than-wt DHFR mutants. The experimental conditions in the paper, however, are likely to be different from those that a wild type strain would face. First, the endogenous promoter of folA regulates the DHFR expression via a negative feedback loop: A drop in DHFR activity/abundance results in the upregulation of its expression (Bershtein, et al., 2015). An interesting question is how the distribution of fitness effects of DHFR mutations will be shaped by the presence of such a regulatory expression element. Second, it was demonstrated that the endogenous DHFR levels in *E. coli* strain carrying the chromosomal folA gene are very close to the optimal level, as the increase in activity or abundance of DHFR does not increase fitness (Bhattacharyya et al., 2016). The fact that over 23% of single point DHFR mutations increase bacterial fitness suggests that the intracellular DHFR levels in the selection system are far away from the optimum. Third, there is no difference in the DHFR sequence between naturally occurring E. coli B and K-12 strains, even though according to the authors' conclusions, the lack of Lon protease in B strains should have driven the adaptive evolution of DHFR in this strain. It would be helpful if the authors discussed these caveats in the manuscript.

We agree that our selection conditions are different from those wild-type DHFR would face in naturally occurring *E. coli* strains. To discuss these caveats, we have made several changes to the manuscript as described below.

To discuss differences in DHFR abundance, we added new data (Figure 1—figure supplement 2) quantifying DHFR abundance of WT DHFR in our selection strain compared to the endogenous DHFR levels in the parent strain:

“As the basis for our studies, we first sought to establish highly sensitive selection conditions for DHFR function that would be calibrated to DHFR enzymatic velocity (rate of DHF conversion per molecule of DHFR) and capable of resolving mutants with velocities near-to or faster-than wild-type. […] We used an *E. coli* strain derived from ER2566 with the genes for DHFR and a downstream enzyme, thymidylate synthase, deleted in the genome and complemented on a pACYC-DUET plasmid with a weak ribosome binding site (see Materials and methods) that results in DHFR abundance at approximately 10% of the endogenous protein level (Figure 1—figure supplement 2, Figure 1—source data 1)”.

We also added a section in the Discussion that addresses differences to selection on naturally occurring DHFR sequences and mentions feedback mechanisms:

“The large fraction of advantageous mutations to DHFR appears to conflict with the fixation of the wild-type DHFR sequence during evolution. […] Nevertheless, our engineered selection conditions yielded considerable insights into constraints on mutational landscapes that are typically hidden from observation precisely because of buffering effects in natural contexts.”